# ChessQA: Evaluating Large Language Models for Chess Understanding

## Abstract

Chess has played an important historical role in AI development, as it has well-defined structure and objective ground truth while admitting a wide spectrum of skill levels. However, existing evaluations of LLM ability in chess are ad hoc and narrow in scope, making it difficult to accurately measure LLM chess understanding and how it varies with scale, post-training methodologies, or architecture choices. We present ChessQA, a comprehensive benchmark that assesses LLM chess understanding across five task categories (Structural, Motifs, Short Tactics, Position Judgment, and Semantic), which approximately correspond to the ascending abstractions that master as they accumulate chess knowledge, from understanding basic rules and learning tactical motifs to correctly calculating tactics, evaluating positions, and semantically describing high-level concepts. In this way, ChessQA captures a more comprehensive picture of chess ability and understanding, going significantly beyond the simple move quality evaluations done previously, and offers a controlled, consistent setting for diagnosis and comparison. Furthermore, ChessQA is inherently dynamic, with prompts, answer keys, and construction scripts that can evolve as models improve. Evaluating a range of contemporary LLMs, we find persistent weaknesses across all five categories and provide results and error analyses by category. We will release the *code*, periodically refreshed *datasets*, and a public leaderboard to support further research.

## 1 Introduction

Since even before the term "artificial intelligence" was coined, chess has been a central yardstick for measuring our progress towards building machines that think. In the 1950s, Shannon and Turing wrote seminal papers on how computers might be programmed to play chess, and in the decades that followed researchers tracked how artificial intelligence was progressing by pitting computer chess programs against increasingly strong human opponents (Shannon, 1950; Turing, 1953). IBM's Deep Blue defeated the reigning World Champion, Garry Kasparov, in 1997, ushering in a period where bespoke models achieved superhuman ability in chess and measuring AI progress with chess was less common (Campbell et al., 2002). But with the rise of generative artificial intelligence and large language models (LLMs), chess has returned as a useful barometer for assessing the depth and quality of machine thinking. How good is the latest LLM, and how good is its reasoning? In a perhaps surprising return to decades-old practice, we are using chess to find out.

Assessing LLM capabilities is essential to AI progress. Generative AI is rapidly transforming industries, the economy, and life in general by providing access to flexible and general intelligence that is broadly useful across domains and needs. The trajectory of this transformation is largely guided by model capability, thus accurately measuring it can help us understand the dynamics of the fundamental shifts taking place. Furthermore, it is equally important to understand model limitations, to support appropriate reliance on LLM outputs and to suggest directions for future improvement. Chess has several desirable properties as a domain for assessing LLM capabilities. It is a neatly circumscribed, perfect-information, and fully objective domain with clear states, actions, goals, and rewards. Despite this, it is incredibly deep, and admits a huge variety of skill levels. People are also still very active in playing chess, and a near-infinite supply of human chess data is available online. Importantly, chess remains anecdotally difficult for LLMs, with contemporary state-of-the-art models routinely failing to complete even simple chess tasks.

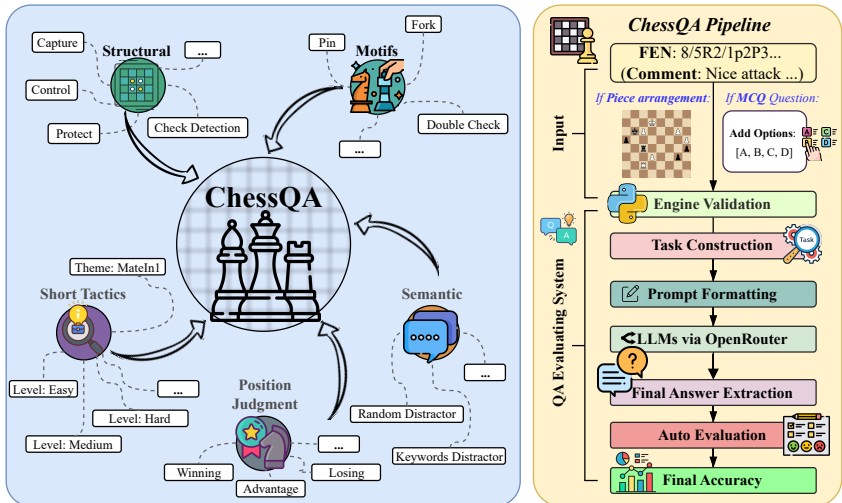

Figure 1: ChessQA at a glance.

However, using chess to assess LLM capabilities is difficult. The sheer vastness of chess means it is not straightforward to actually measure how LLMs do across the entire domain. Most existing ways of using chess to assess models are ad hoc and narrow, often only measuring move quality, performance, or perhaps a narrow set of concepts.

We introduce ChessQA, a unified, comprehensive benchmark that assesses LLM chess understanding across five increasingly complex categories, covering approximately the increasingly abstract levels of understanding that players master as they accumulate chess knowledge. These task categories are Structural (basic rules), Motifs (recurring patterns), Short Tactics (short, targeted calculations), Position Judgment (position evaluation), and Semantic (describing essential high-level concepts in a given position). By constructing a benchmark this way, we capture a more comprehensive picture of chess ability and understanding, going significantly beyond the simple move quality evaluations done previously, and offer a controlled, consistent setting for diagnosis and comparison. Furthermore, ChessQA is inherently dynamic, with prompts, tasks, and construction methods that can evolve as models improve. Evaluating a range of contemporary LLMs, we find persistent weaknesses across all five categories and provide results and error analyses by category.

## 2 RELATED WORK

**Frontier LLMs.** Early general-purpose LLMs scaled decoder-only transformers and were then instruction-tuned and aligned for daily conversations. Contemporary LLMs further scale the model, enable longer context windows, integrate agentic and multi-modal capabilities, and use reinforcement learning for better alignment. Representative proprietary model series include GPT (OpenAI, 2024c; 2025c;d;a; 2024a;b; 2025b), Claude (Anthropic, 2024; 2025a;b), Gemini (DeepMind, 2023; 2024a;b; Comanici et al., 2025), and DeepSeek (DeepSeek-AI, 2024). Popular open-source models include LLaMA series (Grattafiori et al., 2024; Touvron et al., 2023), Qwen series (Yang et al., 2025), Mistral models (team, 2024), and Gemma (Team et al., 2024; 2025).

**LLM evaluation via QA and reasoning.** Question answering (QA) benchmarks have evolved from extractive span selection on short passages (SQuAD; SQuAD 2.0) to open-domain pipelines that learn to retrieve and generate (MS MARCO, Natural Questions, ORQA, REALM, TravelDest) (Rajpurkar et al., 2016; 2018; Bajaj et al., 2016; Kwiatkowski et al., 2019; Lee et al., 2019; Guu et al., 2020; Wen et al., 2024; 2025). To probe multi-step reasoning and robustness, compositional and domain QA introduced multi-hop aggregation, distractors, and truthfulness (HotpotQA, MultiRC, MuSiQue, AmbigQA; ARC, QASC, OpenBookQA, SciQ, CommonsenseQA, BoolQ; DROP; TruthfulQA) (Yang et al., 2018; Khashabi et al., 2018; Trivedi et al., 2022; Min et al., 2020; Clark et al., 2018; Khot et al., 2020; Mihaylov et al., 2018; Johannes Welbl, 2017; Talmor

Table 1: **Positioning of *ChessQA*.** Prior work typically targets a single facet (move selection, state tracking, commentary, gameplay, or multi-game competitions). *ChessQA* unifies five categories with verifiable scoring: *Structural*, *Motifs*, *Short Tactics*, *Position Judgment*, and *Semantic*.

| | Primary focus | Input → Output | Primary metric(s) | Response type | ChessQA mapping |
|---|---|---|---|---|---|
| MATE (Wang et al., 2024a) | Move selection with strategy/tactic rationales | FEN + candidate moves → best move (+ explanation) | Top-1 over candidates; explanation quality | MCQ + free-form rationale | *Short Tactics* |
| PGN2FEN (Cooper, 2025) | State tracking | PGN/SAN sequence → FEN | Exact-FEN; legality gating | Free-form FEN string | *Structural* |
| LLMs on the Chessboard (Kuo et al., 2023) | Formal encodings & compositional constraints | Board encodings/queries → consistency/validity judgments | Accuracy / consistency | Constrained text | *Structural*; *Motifs* |
| LLM Chess Puzzles (Kagi Search, 2025) | Single-move best-move accuracy | FEN → SAN move | Top-1 accuracy (by difficulty) | Free-form SAN | *Short Tactics* |
| LLM Chess Leaderboard (Saplin, 2025) | Text-only gameplay strength | Dialogue (text) moves → full game vs. engine | Elo vs. calibrated engine; win rate (%) | Open-ended dialogue (agent play) | *Short Tactics* |
| Concept-guided Commentary (Kim et al., 2024) | Commentary generation & evaluation | FEN + text → commentary / score | Task-specific protocol | MCQ or scored free-form commentary | *Semantic* |
| Kaggle Game Arena (Lee et al., 2025) | Head-to-head *game* evaluation | Game API/dialogue → complete games | Game outcome | Chess moves | *Short Tactics*; *Position Judgment* |

et al., 2018; Clark et al., 2019; Dua et al., 2019; Lin et al., 2021). As general LLMs saturated earlier sets, exam-style and expert-curated evaluations (MMLU, MMLU-Pro, GPQA, AGIEval) and math benchmarks (GSM8K, MATH, OlympiadBench, Omni-MATH) emphasized harder, diagnostic items (Hendrycks et al., 2020; Wang et al., 2024b; Rein et al., 2024; Zhong et al., 2023; Cobbe et al., 2021; Hendrycks et al., 2021; He et al., 2024; Gao et al., 2024). Long-context stress tests (QuALITY, QASPER, LongBench, RULER) target scaling limits in context use (Pang et al., 2021; Dasigi et al., 2021; Bai et al., 2023; Hsieh et al., 2024).

**Chess AI.** Modern chess engines couple deep evaluation with powerful search (AlphaZero; Leela Chess Zero; NNUE-augmented Stockfish) (Silver et al., 2017; 2018; Leela Chess Zero Project, 2025; Stockfish Developers, 2020b; 2025; Nasu, 2018). Parallel works model *human* decision making in chess (Maia; Maia-2; Maia4All) (McIlroy-Young et al., 2020; Tang et al., 2024; 2025a). On the language side, prior efforts curate commentary corpora and study generating or scoring analyses, often with engine signals in the loop (Jhamtani et al., 2018; Zang et al., 2019; Lee et al., 2022; Feng et al., 2023; Kim et al., 2024).

**Chess-specific LLM evaluations.** Recent efforts evaluate language models on narrow chess sub-skills. **Move-selection** corpora such as MATE curate large position sets with expert-annotated strategic/tactical rationales and report fine-tuned models outperforming proprietary baselines on *select-the-best-move* tasks (Wang et al., 2024a). **State-tracking** benchmarks like PGN2FEN stress whether models can convert move sequences into exact board states (Cooper, 2025). Complementary work probes formal board encodings and compositional reasoning on the chessboard, testing understanding of coordinates, piece relations, and logical constraints over symbolic encodings (Kuo et al., 2023). Beyond chess, (Topsakal et al., 2024) introduce grid-based game competitions and a leaderboard across multiple games that emphasize state tracking and control under agentic play. Community **puzzle/best-move** leaderboards evaluate Top-1 accuracy on single-move positions (Kagi Search, 2025), while **text-only gameplay** leaderboards estimate Elo by playing engines at calibrated levels (Elo, 1978; Saplin, 2025). Beyond move choice, **commentary evaluation** benchmarks introduce concept-guided generation and scoring protocols that align textual analysis with engine signals (Kim et al., 2024). We also note the Kaggle Game Arena, which provides large-scale head-to-head *gameplay* leaderboards and serves as a complementary, agentic lens rather than a QA benchmark (Lee et al., 2025). Table 1 situates representative efforts alongside ours. Orthogonally, **cross-modal isomorphic** evaluations such as SEAM and ISOBENCH include

chess tasks (e.g., FEN strings vs. board images) to test representation-invariant reasoning and reveal modality gaps (Tang et al., 2025b; Fu et al., 2024).

# 3 CHESSQA

We introduce **ChessQA**, a 50-task, 3,500-item benchmark that systematically probes LLM chess understanding along a curriculum of five categories: 11 *Structural* tasks, 6 *Motifs* tasks, 24 *Short Tactics* tasks, 5 *Position Judgment* tasks, and 4 *Semantic* tasks (the distribution of tasks is shown in Figure 2). Going beyond ad hoc testing of move quality or narrow semantic categories, ChessQA comprehensively includes rules knowledge, pattern recognition, concrete calculation, evaluative judgment, and high-level semantics in a controlled setting. In what follows, we outline the benchmark's design principles, specify the prerequisites, and detail breakdown of each category. For readability we place a few representative items (one per category) with their ground truth in Appendix D; full task specifications and grading rules remain in the following section and Appendix C.

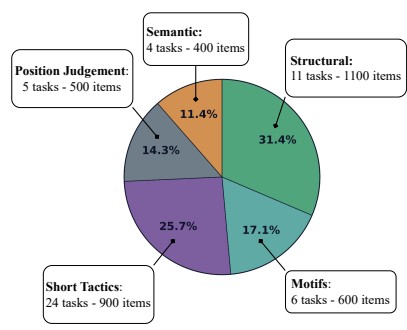

Figure 2: Task distribution in ChessQA.

## 3.1 DESIGN PRINCIPLES

**Comprehensive dimension coverage.** Ideally, a benchmark that measures chess understanding in LLMs should test two fundamental dimensions. (i)It should rigorously measure chess concepts, it must reflect what players actually use over the board: understanding rules and concrete board states; recognizing recurring patterns and motifs; carrying out short, local calculations; judging positional factors; and articulating clear, instructive explanations. (ii) Comprehensive LLM ability evaluation, it must exercise comprehensive reasoning: from fast, short-term checks (legality, immediate tactics, etc.) to slower, long-term judgment like forecasting the trajectory of advantage via bucketed centipawn outcomes and selecting the most informative commentary among plausible alternatives. With this target in mind, **ChessQA** instantiates both dimensions: we sample openings, middlegames, and endgames; include both sides to move; and balance tactical and positional positions. Each item is crafted either to elicit quick calculation or to demand sustained evaluation and explanation, so measured performance tracks genuine chess understanding rather than recall of a few memorized patterns.

**Spectrum of abstraction levels.** Mastering chess, like other subjects, involves understanding concepts and developing knowledge at progressively deeper and more complex levels of abstraction. We design tasks that span five increasingly complex categories that mirror how people typically learn chess: first they learn the rules, recognize patterns, then calculate short lines, then judge positions, and finally explain ideas. Concretely, *Structural* tests for board awareness and single-step legality; *Motifs* requires naming or spotting a chess motif that doesn't require long calculation; *Short Tactics* pushes tests the model's ability to solve tactical puzzles derived from real games; *Position Judgment* asks for an overall quantitative assessment that approximates a chess engine's evaluation; and *Semantic* requires the model to select the most relevant and insightful natural-language comment describing the position from a set of possible comments. This comprises a natural progression in understanding and ability from recognition to reasoning to explanation.

**Calibrated task difficulty.** Ideally, a chess benchmark should have calibrated, controllable difficulty so that we can test models at various levels of complexity. We use two principled levers to achieve this. **(i) Data selection:** we raise difficulty by sourcing tougher positions, e.g., in our *Short Tactics*, we can choose higher-rated tactical puzzles (that have earned their high ratings by being hard to solve in online puzzle solving against humans) so the underlying chess task is more difficult without changing the task format. **(ii) Option design:** with the position fixed, we can

make the answer choice set more demanding. In our *Position Judgment* tasks, we can tighten centipawn granularity by shrinking the bucket grid from a coarse $\{-400, -200, 0, 200, 400\}$ to a finer $\{-200, -100, 0, 100, 200\}$, to modulate the difficulty of selecting the correct answer. Also in our *Semantic* tasks, we control difficulty by varying the method by which we select "distractor" incorrect multiple-choice answers via retrieval: we find comments that are closer to the truth yet objectively wrong, forcing deeper understanding. All of these knobs are parameterized and versioned so that our benchmark scores will remain comparable across future releases as the benchmark keeps pace with rapidly improving LLMs.

## 3.2 PREREQUISITES

**Notations.** Chess is equipped with standardized textual representations: board positions are written in Forsyth–Edwards Notation (FEN), complete games in Portable Game Notation (PGN), and individual moves in either Universal Chess Interface (UCI) or Standard Algebraic Notation (SAN). We use only these notations in prompts and answers. For moves, we choose UCI over SAN because UCI is deterministic at the string level: the source square, target square, and (if any) promotion piece uniquely specify the move and do not rely on board state for disambiguation. This avoids SAN's position-dependent variants (e.g., file/rank disambiguators and check/mate suffixes) and makes parsing and exact-match scoring straightforward. Since FEN/PGN/UCI are ubiquitous in online databases, engines, and forums, models have almost certainly seen them during pretraining and instruction tuning. Using these standards enables a fair zero-shot evaluation without introducing ad hoc formats. (Edwards, 1994; Wikipedia contributors, 2025; Fiekas, 2024a; python chess, 2024)

**Data Sources.** We use data sources that cover broad chess knowledge and provide reliable, objective targets. The Lichess Puzzles corpus (Team, 2025b) is a massive set of puzzles that people continuously train on, and each puzzle is annotated with an empirically-derived rating describing its difficulty and theme tags that describe the motifs involved in the puzzle. The Lichess Evaluations release (Team, 2025a) is a set of aggregated Stockfish analyses for millions of positions at varying depths, supplying centipawn targets and principal variations for each position. Both Lichess puzzles and evaluations databases are continually updated monthly, which is a major advantage for benchmarks such as ours because it allows us to continually refresh slices of data that post-date LLM training cutoffs. For natural-language understanding, we draw expert commentary from ChessBase 17 (ChessBase GmbH, 2023), a human-annotated game database that provides semantically rich explanations. Since ChessBase 17 is a proprietary database, it is unlikely to appear in open web-scale training corpora, thus further preventing data contamination.

**Tools.** We pair reliable chess tooling with scalable retrieval to construct the benchmark. We use python-chess (Fiekas, 2024a) to parse and validate FEN/PGN, enforce move legality, normalize all moves to UCI, and orchestrate engine calls. Stockfish (Stockfish Developers, 2020a; 2025) supplies ground-truth centipawn evaluations and principal variations at controlled depths. For retrieval-based distractors in *Semantic*, we index candidate commentary with FAISS (Johnson et al., 2017) and dense embeddings, allowing us to retrieve semantically close yet incorrect options and to increase difficulty at will.

## 3.3 BENCHMARK CONSTRUCTION

*Structural*: **Basic chess rule understanding.** The first category (11 tasks, 1100 items) measures basic rule-level competence, such as recognizing legal moves, checks, attacked/controlled/protected squares, piece locations, and deterministic board state updates that don't require search or in-depth evaluation. We construct 11 tasks with data derived from public Lichess positions and games, and categorize them into three subtasks: board-state recognition (read what is currently on the board), single-ply[1] (select or validate one legal move), and state updates (apply move sequences and output the board state corresponding to the exact resulting position). Details are shown in Table 2. We also enforce simple legality constraints in the generators: e.g., a piece pinned to its King cannot control or protect squares, and "protect" excludes the king as a target. More details about task construction and the pesudocode are presented in Appendix C.1.

---

[1]In chess, a *ply* is one half-move, i.e. one move made by a single side, either White or Black. A full move is two plies: one half-move by White and one half-move by Black.

*Motifs*: **Chess motifs recognition.** Beyond basic rules, the next step on the path to chess fluency is recognizing tactical motifs and the forcing moves that create them. The *Motifs* category of ChessQA includes 6 tasks and 600 items, which divides motifs into two sub-categories: (1) static motifs (pins, skewers, batteries, and fork squares), which can be identified directly from board position without move simulation, and (2) dynamic motifs, which emerge only through one-ply lookahead search. We input FEN notation of chess positions for all the tasks in this category, and the expected outputs are detailed in Table 3.

Table 2: *Structural* tasks overview.

| Subtask | Input | Output |
|---|---|---|
| **Board-State Recognition** | | |
| Piece arrangement | FEN | Piece list |
| Check detection | FEN | Pieces on square |
| Capture squares | FEN + piece on square | Squares |
| Control squares | FEN + piece on square | Squares |
| Protect squares | FEN + piece on square | Squares |
| **Single-Ply** | | |
| Legal moves (piece) | FEN + target square | UCI moves |
| Legal moves (all) | FEN | UCI moves |
| Check-in-1 | FEN | UCI moves |
| **State Updates** | | |
| State tracking—short | start FEN + 1–5 UCI | FEN |
| State tracking—mid | start FEN + 6–10 UCI | FEN |
| State tracking—long | start FEN + 11–15 UCI | FEN |

*Short Tactics*: **Chess tactic implementation.** Once a model can handle rules and common motifs, the next step is to apply tactics to find the best move in positions. While regular game positions provide diverse data, chess puzzles form a carefully crafted subset of regular positions with a unique best move that typically leads to a decisive advantage. Such a property of chess puzzles enables us to find an absolute best move as ground truth for a given position, instead of distinguishing between multiple moves of

Table 3: *Motifs* subtasks overview

| Subtask | Output |
|---|---|
| Pins | `pinning>pinned>target` |
| Skewers | `attacker>front>back` |
| Forks | `forking>sq1-sq2(-sq3...)` |
| Batteries | `square>square(>square...)` |
| Discovered checks | UCI moves |
| Double checks | UCI moves |

similar quality. Furthermore, puzzles are specifically designed to isolate and teach important tactical patterns and strategic concepts, which are particularly suitable for evaluating an LLM's ability to execute short tactics. Therefore, we choose to use chess puzzles from Lichess Database [2] to construct samples for this category. This category contains 24 subtasks and 900 items, organized according to empirical difficulty recorded by Lichess and the themes of the tactics involved. We measure the first move correctness instead of requiring the LLMs to finish predicting the entire principal variation. We only include puzzles for which the ground truth principal variations are under 6 ply, so as to keep the planning required for LLMs relatively short. Details about subtasks included are reported in Appendix C.3.

*Position Judgment*: **Long-term evaluation.** Once an agent has achieved fluency in how to combine motifs into tactical sequences, the next level of chess understanding is evaluating an entire position. To accurately predict an engine's centipawn advantage (cp) evaluation of a position requires long-horizon strategic reasoning and multi-step planning, making it an ideal task category in our ChessQA benchmark for assessing models' capacity for extended chain-of-thought reasoning and long-term positional understanding in chess. Specifically, centipawn advantage is a chess evaluation metric that quantifies positional superiority in hundredths of a pawn, where +100 represents a material or positional advantage equivalent to being one pawn ahead for White. Questions in this category ask the model to select Stockfish centipawn evaluation in the current position from a set of possible options. For each position $s$, we take the deepest available engine evaluation record from Lichess Evals [3] to determine the ground truth centipawn advantage of $s$, excluding positions where checkmate can be forced for either side. We construct 5 evaluation subtasks by sampling 100 positions for each of five distinct centipawn advantage ranges: Losing (-400±50 cp), Disadvantageous (-200±50 cp), Neutral (0±50 cp), Advantageous (200±50 cp), and Winning (400±50 cp), all evaluated from White's perspective, yielding 500 instances across the full spectrum of positional evaluation. We always give the options as $\{-400, -200, 0, 200, 400\}$ for LLMs to select from. The choice of which exact centipawn options to give is necessarily arbitrary, but these values in particular are aligned with common practice in human chess annotation: "clearly worse" (-400), "slightly worse" (-200), "equal" (0), "slightly better" (200), and "clearly better" (400) correspond to standard Numeric Annotation Glyph categories (e.g., "$\pm$", "$+/-$") that are widely used by chess professionals and enthusiasts to qualitatively evaluate positions. Modern engines expose centipawn scores that chess practitioners routinely map to these discrete bands when annotating games. For ex-

---

[2]https://database.lichess.org/#puzzles

[3]https://database.lichess.org/#evals

ample, Stockfish's NNUE evaluation (Stockfish Developers, 2020b) and the Stockfish 17.1 release notes (Stockfish Developers, 2025) are representative examples of such usage, as is the Lichess Evaluations database (Team, 2025b;a), which aggregates engine scores for millions of positions that players mentally convert to corresponding qualitative labels in practice. Unlike previous categories that are testing chess understanding with isolated static features or short dynamics, centipawn evaluation requires long-term look-ahead and integrating multiple positional dimensions into a holistic assessment that reflects both immediate tactics and strategic planning.

*Semantic*: **Chess-specific language understanding.** Beyond choosing moves and making judgments, arguably the highest abstraction level of chess mastery is the ability to talk fluently about a position, and explain one's reasoning. A capable model should be able to connect a concrete move and position to the strategic and tactical ideas at play. This category (4 tasks, 400 items) evaluates grounded natural-language understanding tied to a specific move and position. Each item provides the FEN, the just-played UCI move; four human comments are shown, exactly one of which is true, specific, and informative for that context. Distractors create lexical, structural, and semantic confounds. The model answers with a single letter (A/B/C/D). Comment text is anonymized and normalized (e.g., player names are replaced with the color they are playing); a strict position-truth judge admits only comments whose claims are supported by the paired move and position. The *random* subtask serves as a baseline with i.i.d. sampled distractors, while three confound-controlled subtasks gradually introduce more difficulty: *keyword* introduces lexical confusion by selecting distractors sharing chess terminology (e.g., fork, safety), *piece_stage* creates structural ambiguity using comments from positions with identical piece types and game phases, and *embedding* finds distractors through dense retrieval of similar comments using Qwen3-Embedding-8B (Zhang et al., 2025). This progression from random to semantically similar distractors enables fine-grained assessment of models' chess-specific language understanding beyond pattern matching. More details are reported in Appendix C.

## 4 EXPERIMENTS

We conduct extensive experiments to evaluate the chess understanding capabilities of frontier LLMs on our proposed ChessQA benchmark. We aim to probe the general and per-category performance of LLMs as well as their token- and cost-effectiveness.

### 4.1 EXPERIMENTAL SETTINGS

We evaluate 15 contemporary LLMs spanning 7 families on ChessQA, including Anthropic, DeepSeek, Google, Meta, Mistral AI, OpenAI, and Qwen. Models are treated strictly as black-box text-to-text models without agentic behavior, such as tool use, online search, and code execution. We enable thinking with medium efforts if the model supports thinking mode, allowing at most 32K tokens to be generated. Other non-thinking models are given 8K tokens as the budget to generate responses. We ran GPT-5, Gemini 2.5 Pro/Flash, Claude Sonnet 4/4.5, Claude Haiku 4.5, DeepSeek V3.1, and Qwen3 Next 80B with both thinking and non-thinking modes to investigate the effectiveness of generating long Chain-of-Thought reasoning chains for chess understanding, resulting in 23 runs in total. We run all models under their default sampling parameters for generation using Open-Router (OpenRouter, 2023). To clarify GPT-5 specifically, GPT-5 Chat is the non-thinking version on OpenRouter, and we use GPT-5 to refer to this version of GPT-5 Chat in this paper. We extract the final answer with pre-defined patterns that have been added to the prompts and use exact matching to evaluate the correctness of each response. Specifically, for questions that require multiple answers (e.g., find all legal moves), we first parse the extracted final answer into a set, and then apply exact set matching to evaluate its correctness. This design avoids extraneous errors introduced in answer processing, enabling us to better isolate and measure the chess understanding capabilities of LLMs. In what follows, we add a * after the model name to denote that thinking is enabled.

### 4.2 RESULTS

**Overall performance.** As shown in Fig 3, most models achieve relatively low performance on our ChessQA benchmark, with only 4 runs achieving over 50%. GPT-5* with thinking is the best performing model, achieving an accuracy of 79.3%. Unlike in many open benchmarks (Wang et al.,

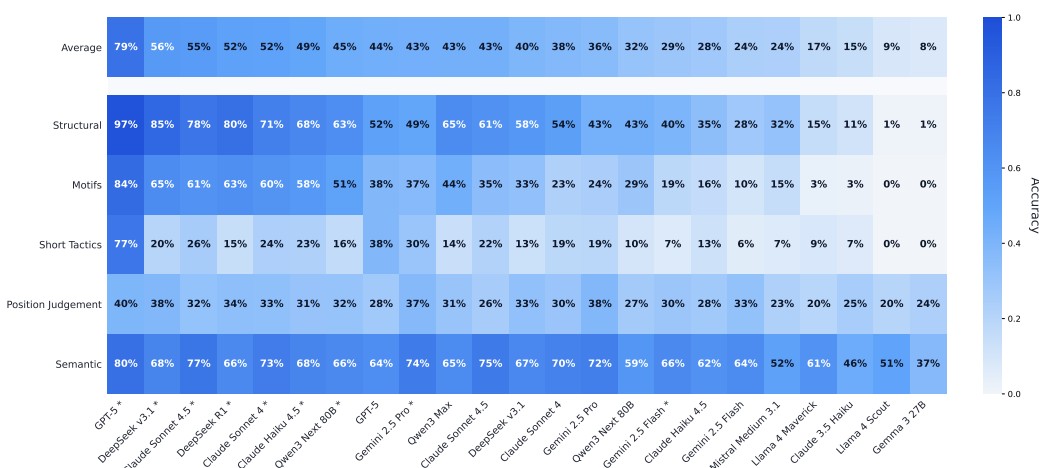

Figure 3: The overall and per-category performance comparison. * denotes thinking enabled.

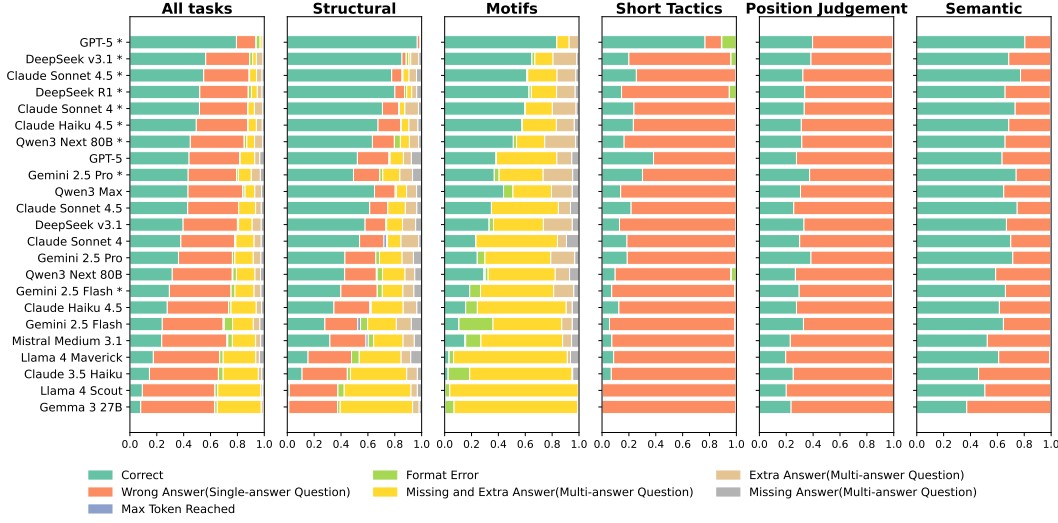

Figure 4: Breakdown of response evaluation results.

2024b) where proprietary models significantly outperform open-source alternatives in general, we found state-of-the-art open-source models such as DeepSeek v3.1* and Qwen3 Next 80B* ranking high in our leaderboard. Performance consistently follows scaling laws across both proprietary and open-source model families, with larger variants outperforming their smaller counterparts (Gemini-2.5-Pro* at 43% vs. Gemini-2.5-Flash* at 29%; Llama 4 Maverick at 17% vs. Scout at 9%), highlighting the impact of model scale on chess understanding capabilities.

**Performance across categories.** As shown in Fig 3, model performance differs significantly across categories, even the best performing GPT-5* with thinking that can achieve 97% in the empirically easiest *Structural* tasks, can only achieve 40% in *Position Judgment*. Notably, *Short Tactics* represent the hardest category (mean 17.4%) and most models fall under 20%. Also, *Position Judgment* accuracy remains stubbornly limited even among top-performing models. Since *Position Judgment* are essentially 5-way classification problems, a random guess will theoretically get 20%. These two categories, which involves short and long-term look-ahead in chess, appear to be more challenging for LLMs than other static pattern recognition categories, demonstrating their

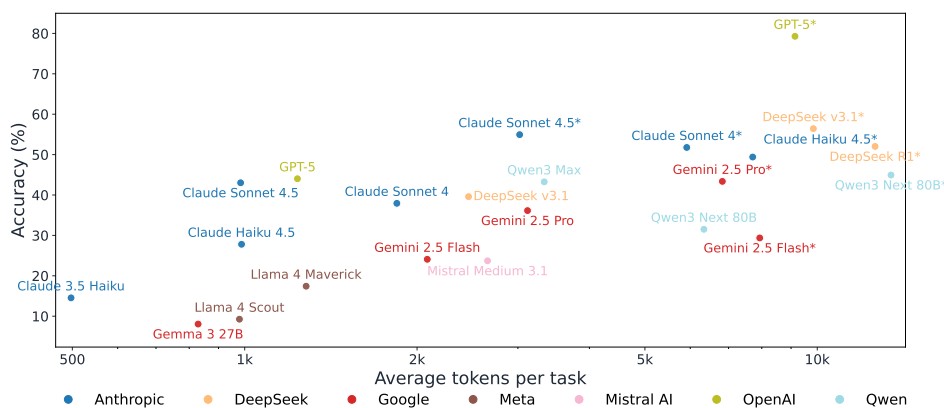

Figure 5: Performance comparison w.r.t #tokens per problem. * denotes thinking enabled.

deficiency in implicit search and long-term planning. However, although GPT-5* performs soundly on our tasks, it still has room for improvement. We crafted a set of 100 high-rated Lichess puzzles (rating above 3000) and evaluated GPT-5* on this set. GPT-5* with high reasoning effort attains **26** out of 100, underscoring that expert-level *Short Tactics* remains challenging for even the very best models (details in Appendix B.4).

**Reasoning efforts and token efficiency.** Our analysis highlights the substantial impact of reasoning efforts: models employing explicit reasoning consistently outperform their base counterparts. Pairwise comparisons indicate an average accuracy improvement of **+14.7** percentage points when reasoning is activated (e.g., GPT-5 improved from 44.0% to 79.3%, DeepSeek v3.1 from 39.6% to 56.4%, and Claude Sonnet 4 from 41.7% to 51.8%). From Fig 5, we analyzed token utilization across all models, observing significant variability. The number of tokens employed per task varied significantly, ranging from approximately 497 tokens to over 13,452 tokens. Models using explicit reasoning strategies consistently consumed more tokens (9,142 tokens on average for GPT-5*) compared to their non-reasoning counterparts (1,236.31 tokens on average for GPT-5). Notably, as shown in Appendix Fig 7, higher token usage correlated positively with accuracy improvements across nearly all categories, suggesting the effectiveness of test-time scaling. However, we question the token efficiency of models as human chess players won't need an average of 11,668 tokens per problem to fully verbalize the solution to a chess puzzle. Therefore, the token efficiency is an important direction towards better LLMs, in particular for chess understanding.

**Error analysis.** We examined incorrect GPT-5* responses and identified four frequent failure modes: (1) board-state hallucination or incorrect piece/square recognition, (2) legality reasoning errors in short tactical problems, (3) correct analytical reasoning leading to an incorrect final move choice, and (4) false assertions of "no answer." In mode (1), GPT-5* occasionally misreads board states, such as confusing the knight's position or inventing phantom pieces, leading to illegal moves (Appendix E.1). Mode (2) revealed contradictions or flawed heuristics about move legality (e.g., forced move contradictions or misapplication of adjacency heuristics; Appendix E.2). Mode (3) demonstrated instances where sound intermediate reasoning resulted in suboptimal final moves, like overlooking underpromotion advantages or selecting inferior tactical continuations (Appendix E.3). Lastly, mode (4) captures instances where GPT-5* erroneously concluded "no answer," neglecting existing valid solutions due to misapplication of tactics or misinterpretation of positional constraints (Appendix E.4). To verify the existence of the board-state hallucination problem, we explicitly add piece arrangement context into the prompts. As shown in Table 4 in the Appendix, when piece arrangements are given, the overall performance is significantly improved. Such a setting is equivalent to eliminating the difficulty of board-state recognition for LLMs, reflecting the severity of the board-state hallucination problem in LLM reasoning.

**Hyperparameter sensitivity.** To calibrate how much test-time reasoning matters, and whether sampling alone helps, we ran hyperparameter sensitivity experiments with Claude Haiku 4.5. With

thinking enabled, increasing the reasoning budget (via higher max tokens and reasoning effort) yields clear gains: accuracy rises from **28.9%** (8K/low) to **49.4%** (16K/medium) and **54.0%** (32K/high). Item-paired $t$-tests indicate the low-to-medium jump is highly significant; the medium-to-high improvement is smaller but still significant. Without thinking, changing temperature from **1.0** to **0.7** produces no statistically significant difference overall. Full setups, per-category results, and significance tests are in Appendix B.2.

## 5 DISCUSSION

**Chess as measurement tool.** Chess was called the "drosophila of AI" by John McCarthy and others as artificial intelligence was first developed. Bespoke, domain-specific programs eventually surpassed human ability in chess, but with the advent of general artificial intelligence, chess has resumed its role as the ideal testbed for measuring our progress towards developing AI. Chess is perfectly objective with clear states, actions, and rewards, it spans a deep skill gradient with abundant human data, and yet modern LLMs still fail at seemingly simple chess tasks. Existing chess-LLM evaluations are typically ad hoc (best-move, narrow tasks, gameplay), which makes holistic measurement difficult. Our contribution is to unify these perspectives in a controlled QA setting that measures broad understanding and capabilities.

**Takeaways.** Across models and categories we find persistent weaknesses, but explicit reasoning consistently helps: enabling it yields sizeable average gains (+14.7 points on pairwise comparisons), with higher token budgets correlating with higher accuracy, suggesting models effectively leverage additional reasoning tokens. The benchmark is robust to prompt-format perturbations, and adding piece-arrangement context shifts category accuracies in predictable ways. Our qualitative error analysis reveals four recurrent failure families: (1) board-state hallucination/misrecognition; (2) legality mistakes in short tactics; (3) sound analysis but wrong final action; and (4) false "no answer." These patterns indicate that state parsing and action selection remain bottlenecks even when intermediate analysis is plausible.

**Future work.** By spanning rules, patterns, calculation, evaluative judgment, and explanation in a single, verifiable QA framework, ChessQA provides a diagnostic map for intervention. Because the pipeline is parameterized and versioned, and because chess is practically infinite with respect to data, we can periodically refresh evaluation data to maintain perennially-fresh leaderboards, as well as steadily increase difficulty and maintain comparability as models improve. Together, these aspects make ChessQA a sustained, useful measurement suite for LLM chess understanding.

## 6 LIMITATIONS

Our study has limitations inherent to the complexity of chess and the scope of diagnostic evaluation. First, we do not include a human baseline due to limited budget for human experiments; instead, we anchor task difficulty to objective, widely used metrics such as Lichess puzzle ratings. Second, while our category-level sample sizes are substantial, computational costs preclude exhaustive testing across all item permutations, so we report confidence intervals on key aggregates. Third, although ChessQA spans five abstract categories, the concept set could be further extended, for instance, we cap puzzle depth for Short Tactics and utilize coarse bins for Position Judgment—though our parameterized pipeline supports future extensions. Additionally, prices differ across providers and can change over time, especially when using a request routing system such as OpenRouter, making it difficult to measure cost efficiency rigorously. Finally, this work is strictly diagnostic: we provide a verifiable benchmark and leaderboard to track progress, but we do not propose specific training interventions to improve LLM performance.

## 7    ETHICS STATEMENT

This work adheres to the ICLR Code of Ethics. No human subjects or animal experimentation were involved in this study. All datasets utilized, including Lichess puzzle databases and public chess game archives, were obtained in compliance with relevant usage guidelines, ensuring no violation of privacy. We have made concerted efforts to prevent biases and discriminatory outcomes throughout our research process. No personally identifiable information was utilized, nor were any experiments conducted that could compromise privacy or security. We are committed to upholding transparency and integrity throughout the research process.

## 8    REPRODUCIBILITY STATEMENT

We have made significant efforts to ensure the reproducibility of the results presented in this paper. All code and datasets are publicly available through an anonymous repository, enabling replication and verification of our experiments. The experimental setup, including model evaluations, prompt construction, and hardware specifics, is comprehensively detailed within the paper. Additionally, we have provided a thorough description of our contributions, including the design and construction of the ChessQA benchmark, generation and curation of datasets spanning Structural, Motifs, Short Tactics, Position Judgment, and Semantic categories, implementation of evaluation scripts, and extensive analyses of contemporary LLM performance.

Furthermore, public datasets utilized in this work, such as Lichess puzzle sets and FEN/PGN archives from chess databases, are openly accessible, ensuring consistent and reproducible evaluation outcomes. We believe these measures will facilitate the reproduction of our findings and contribute to ongoing advancements in this field.

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

## A    USE OF LARGE LANGUAGE MODELS STATEMENT

Large language models were utilized exclusively for improving the clarity of writing and correcting grammatical errors. They were not employed for generating research ideas or influencing the intellectual substance of this work.

## B    ADDITIONAL RESULTS

### B.1    EXTENDED RESULTS AND DIAGNOSTICS

Table 4: Effect of prompt context on overall accuracy.

|  | Base | $\Delta$Example | +PieceArr |
| --- | --- | --- | --- |
| Claude 3.5 Haiku | 14.5% | 14.5% | **23.5%** |
| Gemini 2.5 Flash | 24.1% | 23.4% | **31.0%** |

**Benchmark Robustness.**    We gave format examples in the prompts in order to extract final answers from LLMs' responses. As shown in Table 4, when a different set of format examples is given ($\Delta Example$), model performance is mostly consistent with the results when the default format examples are used (Base). Such results demonstrate the robustness of ChessQA with regard to variant format examples, and our curated prompts ensure that models do not use format examples as in-context learning examples.

**Cost Effectiveness.**    We evaluated models in terms of accuracy per dollar spent in Fig B.1, highlighting significant disparities driven by varying model architectures and pricing structures. The cost per problem ranged from as low as $0.0002 to as high as $0.09, a difference spanning over 500 times. Gemma 3 27B emerged as the most cost-effective model, achieving approximately 480 accuracy points per dollar, despite its relatively low absolute accuracy. Conversely, top-performing models like GPT-5* demonstrated superior accuracy but at a significantly higher cost, emphasizing a trade-off between raw performance and economic efficiency. It is important to note that these cost evaluations are subject to slight inaccuracies due to routing variability in model queries.

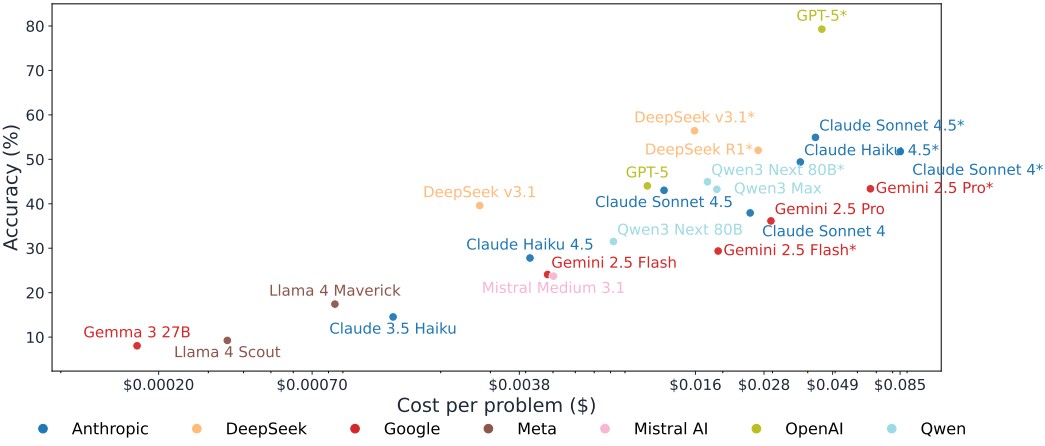

Figure 6: Performance comparison w.r.t cost per problem. * denotes thinking enabled.

**Token Efficiency.**    In Fig 7, we provide detailed insights into token consumption per task across different categories. Reasoning-enabled models consistently utilize significantly more tokens, with GPT-5* notably using up to 14,823 tokens per task in Position Judgment and 11,668 tokens in Short Tactics. Base models generally consume fewer tokens, reflecting a reduced depth in their reasoning

process. Token usage varies substantially by category, with Position Judgment and Short Tactics consistently demanding higher token counts due to their complexity and the need for intricate reasoning paths. Conversely, Structural and Semantic tasks show more modest token usage, reflecting their relative ease and straightforward nature.

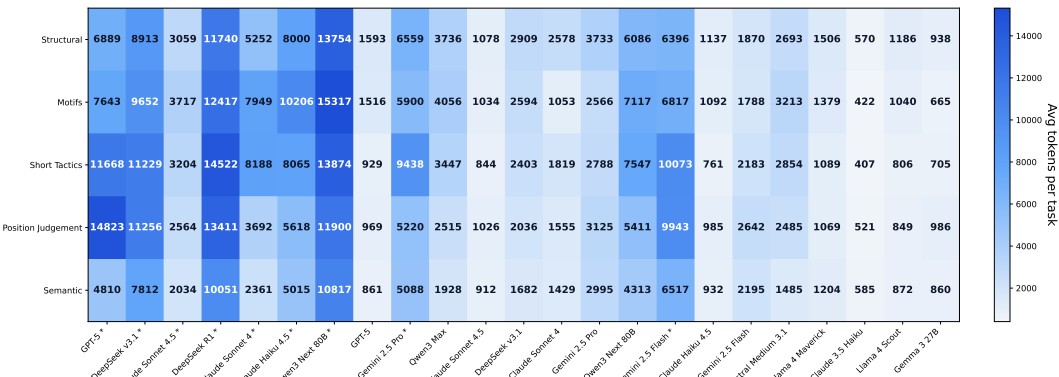

Figure 7: The overall and per-category tokens used comparison. * denotes thinking enabled.

**Model Agreement.** Fig 8 presents cross-model agreement across all evaluated tasks, measured by average accuracy overlap. High-performing models like GPT-5*, DeepSeek v3.1*, and Claude Sonnet 4* exhibit strong mutual agreement (up to 74.6%), indicating similar reasoning strategies and correct solution pathways. Conversely, lower-performing models demonstrate significantly less agreement, often below 50%. The heatmap clearly reveals clusters of higher agreement among top-tier models, suggesting that advanced reasoning capabilities converge on similar solutions, whereas divergence among lower-tier models highlights variability and inconsistency in basic tactical comprehension.

## B.2 HYPERPARAMETER SENSITIVITY

**Reasoning Effort Sensitivity.** We evaluate explicit max token budgets and reasoning effort for Claude Haiku 4.5* across four settings of thinking budgets: 4K (minimal), 8K (low), 16K (medium), and 32K (high). Table 5 summarizes overall accuracy and average token usage, while Table 6 details per-category performance. While minimal and low effort settings perform comparably to the non-thinking baseline ($\approx$28%), the medium and high settings yield substantial gains, achieving 49.4% and 54.0% accuracy, respectively. A paired t-test over the 3,500 items confirms that the high-effort setting significantly outperforms the minimal/low condition (+25.7 percentage points; $t=28.14$, $p<10^{-16}$). The improvement from medium to high effort is also statistically significant, though smaller in magnitude (+4.6 percentage points; $t=5.39$, $p<10^{-6}$). These findings are consistent with established test-time compute scaling laws.

Table 5: Claude Haiku 4.5 reasoning-effort hyperparameter sensitivity results (thinking enabled). Average tokens are per problem.

| Effort | Overall accuracy | Avg tokens / problem |
|--------|------------------|----------------------|
| Minimal | 28.9% | 1,638 |
| Low | 28.9% | 1,507 |
| Medium | 49.4% | 7,713 |
| High | **54.0%** | 12,291 |

**Temperature Sensitivity.** We examine the impact of sampling temperature (1.0 vs. 0.7) on the non-thinking Claude Haiku 4.5 model across the full benchmark. Overall accuracy remains stable at 28.2% (temp 1.0) versus 27.7% (temp 0.7). A paired t-test on item-level correctness yields no

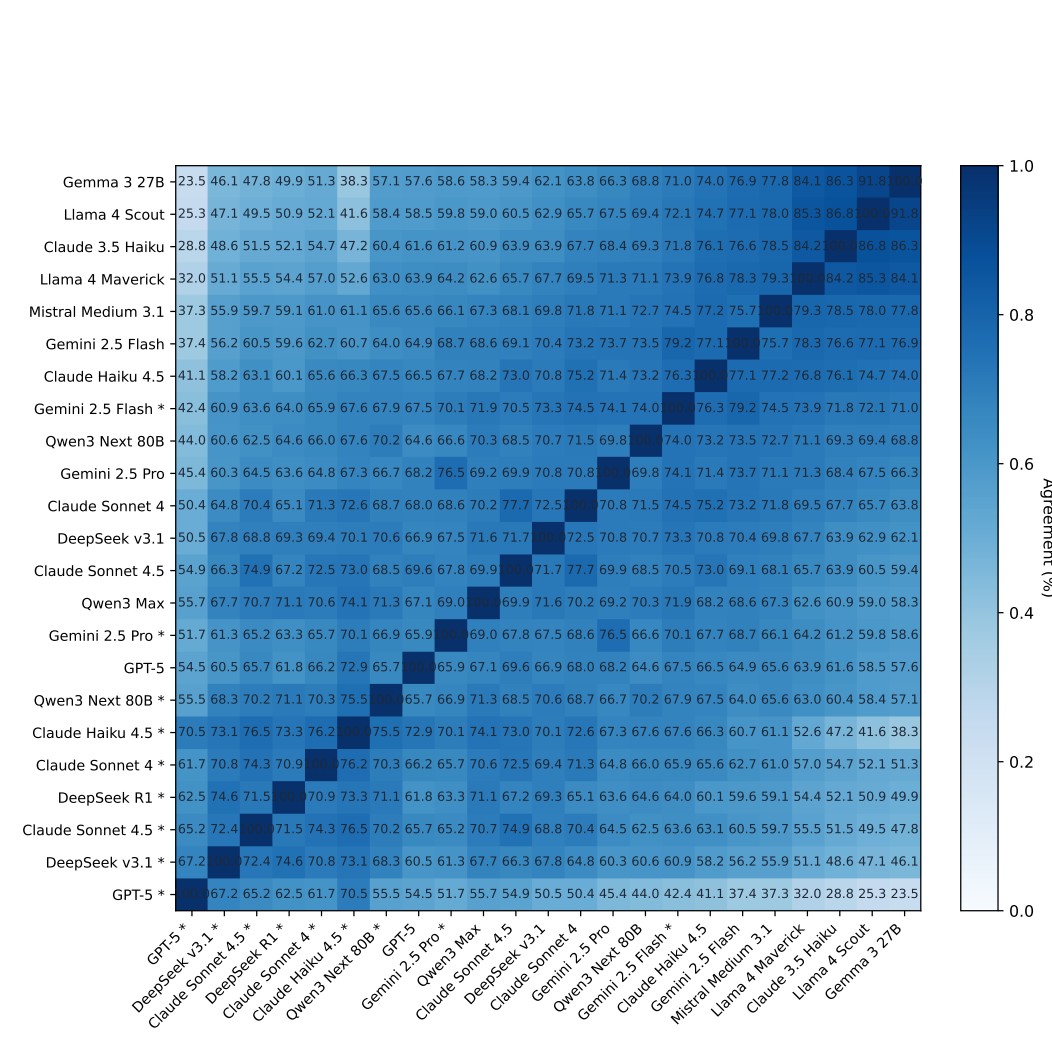

Figure 8: Cross-model agreement on all tasks, shown by average accuracy.

Table 6: Per-category accuracy for Haiku-4.5 across effort settings.

| Category | Minimal | Low | Medium | High |
|---|---|---|---|---|
| Structural | 36.3% | 37.9% | 67.5% | **75.6%** |
| Motifs | 20.8% | 15.0% | 57.7% | **62.5%** |
| Short Tactics | 10.2% | 10.7% | 23.3% | **27.7%** |
| Position Judgement | 28.6% | **31.8%** | 31.2% | 31.2% |
| Semantic | 62.8% | 62.0% | 68.5% | **69.3%** |

statistically significant difference ($t$=0.85, $p$=0.40). Similarly, we observe no significant deviations within individual categories (Motifs $p$=0.92, Position Judgement $p$=0.28, Semantic $p$=0.73, Short Tactics $p$=0.77, Structural $p$=0.55). Table 7 provides the full per-category breakdown.

Table 7: Temperature ablation for Claude Haiku-4.5 (non-thinking).

| Category | Temp 1.0 | Temp 0.7 |
|---|---|---|
| Structural | 35.0% | 34.3% |
| Motifs | 17.0% | 16.8% |
| Short Tactics | 13.6% | 13.2% |
| Position Judgement | 28.0% | 25.8% |
| Semantic | 59.8% | 60.5% |

## B.3 STOCHASTIC VARIATION

To quantify randomness from generation, we ran three independent non-thinking evaluations of Claude Haiku 4.5 at temperature $T$=1.0 and report mean $\pm$ sd accuracy per category and overall (Table 8). Overall variability is small (27.8% $\pm$ 0.4), and category error bars are similarly modest in *Structural* (34.8% $\pm$ 0.2) and *Position Judgment* (27.7% $\pm$ 0.2), somewhat larger in *Motifs* (16.4% $\pm$ 0.7), and largest in *Short Tactics* (12.2% $\pm$ 1.6), reflecting that tactical items are more sensitive to local sampling differences. Importantly, the category ranking and the qualitative conclusions of the paper remain unchanged across runs, indicating that run-to-run randomness does not account for the large gaps we observe between categories.

Table 8: Claude Haiku 4.5: Stochastic variation over three runs. We report accuracy (%) as mean $\pm$ sample standard deviation over three independent runs, with a second column shows the observed min–max range across runs.

| Category | Accuracy (mean $\pm$ std) | Min | Max |
|---|---|---|---|
| Structural | 34.8 $\pm$ 0.2 | 34.6 | 35.0 |
| Motifs | 16.4 $\pm$ 0.7 | 15.7 | 17.0 |
| Short Tactics | 12.2 $\pm$ 1.6 | 10.4 | 13.6 |
| Position Judgment | 27.7 $\pm$ 0.2 | 27.6 | 28.0 |
| Semantic | 60.8 $\pm$ 0.9 | 59.8 | 61.5 |
| Overall | 27.8 $\pm$ 0.4 | 27.4 | 28.2 |

## B.4 HARD PUZZLE SOLVING

We assembled a split of the 100 high-rated Lichess puzzles (rating over 3000) and evaluated GPT-5* with high thinking effort. On this extreme slice, GPT-5* achieved 26 out of 100 (average 24,482 tokens per problem). While absolute accuracy is modest by design, this stress test corroborates the main finding that short tactics remain difficult even for frontier models when difficulty is adversarially high.

## C  Task Design Details

### C.1  *Structural*

**Scope and answer format.**  *Structural* targets rule-level competence only: legal move generation, check relations, attack/control/protect sets, full piece listing, and deterministic state updates from UCI moves. Prompts allow free-form reasoning but require a single terminal line:



FINAL ANSWER: <answer>



We provide type-specific format exemplars inside each prompt to reduce formatting errors; prompts are constructed with fixed format examples at build time. Gold answers are canonicalized at generation (sorted lists, fixed piece-order), enabling exact-match scoring on the terminal line.

**Subtask specifications.**

- **Piece arrangement**: Given a FEN, list all pieces in a fixed order (White: K,Q,R,B,N,P; then Black: K,Q,R,B,N,P). For each piece type, squares are alphabetized.
- **Legal moves (piece/all)**: Enumerate legal UCI moves from the position; the per-piece variant conditions on a target square.
- **Check detection**: If the side to move is in check, list all checking pieces as *"Color Piece at <square>"*.
- **Check-in-1**: List all legal moves that give immediate check.
- **Capture / Control / Protect squares**: For a specified *non-pawn, non-king* piece and square, return, respectively: *enemy-occupied* attacked squares; *empty* attacked squares; and *friendly-occupied* attacked squares *excluding the king*. If the piece is pinned, the set is empty.
- **State tracking (short/mid/long)**: Apply a sequence of UCI moves (lengths 1–5 / 6–10 / 11–15) to a start FEN and output the exact resulting FEN.

**Generation pipeline.**  We build eight puzzle-based subtasks from a shuffled Lichess puzzle CSV, taking at most one subtask per puzzle ID to prevent reuse; state-tracking items are extracted from broadcast PGNs after the first 30 half-moves. Task builders and prompt constructors follow the structural generation script and utilities. Full algorithm showed by Algo. 1.

**Canonicalization and scoring.**

- **Evaluation:** exact string match on the terminal line after extracting the substring following FINAL ANSWER:.
- **Sorted outputs:** multi-item answers are sorted at gold construction (e.g., legal moves, check-in-1, capture/control/protect).
- **Check detection:** checking pieces are returned in the chess library's attacker enumeration order (no additional sort).
- **Arrangement:** piece group order is fixed; per-type square lists are alphabetized.

**Prompts and exemplars.**  We show prompts of the tasks receptively.

**Piece arrangement**

```
You are given a chess position in FEN: <FEN>.
Provide the complete piece arrangement of this position. List all pieces with
↪   their colors, types, and squares.
Format: 'Color PieceType: [square1, square2, ...]', separated by commas and
↪   spaces for different piece types.
List the pieces in the order of White pieces first (King, Queen, Rook, Bishop,
↪   Knight, Pawn) followed by Black pieces (King, Queen, Rook, Bishop, Knight,
↪   Pawn).
```

---

**Algorithm 1** Structural Tasks Generation

---

1: **Inputs:** Lichess puzzle CSV (FEN,PuzzleId), broadcast PGN file
2: **Params:** $N_{sample} = 100$ per subtask; seed $= 42$
3: SEEDEVERYTHING(42); $D \leftarrow$ READPUZZLES(); shuffle once
4: $U \leftarrow \emptyset$                                                                                                         ▷ used puzzle IDs
5: Initialize counters for the 11 subtasks to 0
6: **for** row in $D$ **do**
7:     **if** row.PuzzleId $\in U$ **then continue**
8:     **end if**
9:     $B \leftarrow$ BOARD(row.FEN)
10:     **for** task $\in \{$arrangement, legal_piece, legal_all, check_det, check_in_1, capture, protect, control$\}$ **do**
11:         **if** counter[task] $\geq N_{sample}$ **then continue**
12:         **end if**
13:         $t \leftarrow$ GENERATE_task$(B, \dots)$                                                 ▷ returns NONE if preconditions fail
14:         **if** $t \neq$ NONE **then**
15:             append $t$; counter[task]++; $U \leftarrow U \cup \{$row.PuzzleId$\}$; **break**
16:         **end if**
17:     **end for**
18:     **if** all eight puzzle-based counters $\geq N_{sample}$ **then break**
19:     **end if**
20: **end for**
21: **// State tracking from PGNs**
22: **for** game in READPGN() **do**
23:     **if** all $\{$short, mid, long$\}$ counters $\geq N_{sample}$ **then break**
24:     **end if**
25:     **for** (subtype, min, max) in $\{$(short,1,5),(mid,6,10),(long,11,15)$\}$ **do**
26:         **if** counter[subtype] $\geq N_{sample}$ **then continue**
27:         **end if**
28:         $L \leftarrow$ random integer in $[min, max]$
29:         (start_fen, moves, game_id) $\leftarrow$ EXTRACTFRAGMENT(game, start_after=30, track=$L$)
30:         **if** moves **then**
31:             $t \leftarrow$ MAKESTATETRACKING(game_id, start_fen, moves, subtype)
32:             append $t$; counter[subtype]++; **break**
33:         **end if**
34:     **end for**
35: **end for**
36: SAVETASKS(structural.jsonl)

---

```
If a piece type has no pieces on the board, skip it in the listing.
List the squares for each piece type in alphabetical order.
Analyze step by step and explain your reasoning.
Finish with a single line formatted EXACTLY as:
FINAL ANSWER: <answer>
Example final answer: White King: ['e1'], White Queen: ['d1'], White Rook:
↪  ['a1', 'h1'], White Bishop: ['c1', 'f1'], White Knight: ['b1', 'g1'], White
↪  Pawn: ['a2', 'b2', 'c2', 'd2', 'e2', 'f2', 'g2', 'h2'], Black King: ['e8'],
↪  Black Queen: ['d8'], Black Rook: ['a8', 'h8'], Black Bishop: ['c8', 'f8'],
↪  Black Knight: ['b8', 'g8'], Black Pawn: ['a7', 'b7', 'c7', 'd7', 'e7', 'f7',
↪  'g7', 'h7']
```

**Legal moves (piece)**

```
You are given a chess position in FEN: <FEN>.
Find all legal moves for the piece on square <square>. List the moves in UCI
↪  format, separated by commas and spaces.
Analyze step by step and explain your reasoning.
Finish with a single line formatted EXACTLY as:
FINAL ANSWER: <answer>
Example final answer: e2e3, e2e4
```

**Legal moves (all)**

```
You are given a chess position in FEN: <FEN>.
Find all legal moves in this position. List the moves in UCI format, separated
↪  by commas and spaces.
Analyze step by step and explain your reasoning.
Finish with a single line formatted EXACTLY as:
FINAL ANSWER: <answer>
Example final answer: e2e4, c2b1q
```

**Check detection**

```
You are given a chess position in FEN: <FEN>.
In this position, the side to move is in check. Identify the piece(s) that is
↪  delivering the check.
Analyze step by step and explain your reasoning.
Finish with a single line formatted EXACTLY as:
FINAL ANSWER: <answer>
List each checking piece with its color, type, and square (e.g., White Queen at
↪  e5). Separate multiple pieces with commas and spaces if applicable.
```

**Check-in-1**

```
You are given a chess position in FEN: <FEN>.
Find all moves that put the opponent in check. List the moves in UCI format,
↪  separated by commas and spaces.
Analyze step by step and explain your reasoning.
Finish with a single line formatted EXACTLY as:
FINAL ANSWER: <answer>
Example final answer: e2e4, c2b1q
```

**Capture squares**

```
You are given a chess position in FEN: <FEN>.
Find all squares that the <Color Piece> on <square> can capture (i.e. every
↪  square that has an opponent piece such that the <Color Piece> on <square>
↪  could legally move to that square and capture the piece).
Exclude captures if the <Color Piece> on <square> is pinned to its king and thus
↪  cannot move.
Analyze step by step and explain your reasoning.
Finish with a single line formatted EXACTLY as:
FINAL ANSWER: <answer>
Example final answer: e4, f5
```

**Control squares**

```
You are given a chess position in FEN: <FEN>.
Find all squares that the <Color Piece> on <square> controls (i.e. every empty
↪  square that the <Color Piece> on <square> could legally move to, excluding
↪  squares occupied by any piece).
Exclude control if the <Color Piece> on <square> is pinned to its king and thus
↪  cannot move.
Analyze step by step and explain your reasoning.
Finish with a single line formatted EXACTLY as:
FINAL ANSWER: <answer>
Example final answer: e4, f5
```

**Protect squares**

```
You are given a chess position in FEN: <FEN>.
Find all squares that contain pieces that the <Color Piece> on <square> protects
↪  (i.e. every square that contains a piece such that the <Color Piece> on
↪  <square> could legally recapture if an enemy piece captured it, excluding
↪  the king since it can't be captured).
```

```
1296   Exclude protection if the <Color Piece> on <square> is pinned to its king and
1297   ↪  thus cannot move.
1298   Analyze step by step and explain your reasoning.
1299   Finish with a single line formatted EXACTLY as:
1300   FINAL ANSWER: <answer>
1301   Example final answer: e4, f5
1302
1303   State tracking—short
1304
1305   Given an initial FEN and a sequence of UCI moves, apply the moves in order and
1306   ↪  output the exact resulting FEN.
1307   Initial FEN: <start FEN>
1308   Moves (UCI): <u1 u2 ...>
1309   Analyze step by step and explain your reasoning.
1310   Finish with a single line formatted EXACTLY as:
1311   FINAL ANSWER: <answer>
       Example final answer: rnbqkbnr/pppppppp/8/8/8/8/PPPPPPPP/RNBQKBNR w KQkq - 0 1
1312
1313   State tracking—mid
1314
1315   Given an initial FEN and a sequence of UCI moves, apply the moves in order and
1316   ↪  output the exact resulting FEN.
1317   Initial FEN: <start FEN>
1318   Moves (UCI): <u1 u2 ...>
1319   Analyze step by step and explain your reasoning.
1320   Finish with a single line formatted EXACTLY as:
1321   FINAL ANSWER: <answer>
       Example final answer: rnbqkbnr/pppppppp/8/8/8/8/PPPPPPPP/RNBQKBNR w KQkq - 0 1
1322
1323   State tracking—long
1324
1325   Given an initial FEN and a sequence of UCI moves, apply the moves in order and
1326   ↪  output the exact resulting FEN.
1327   Initial FEN: <start FEN>
1328   Moves (UCI): <u1 u2 ...>
1329   Analyze step by step and explain your reasoning.
1330   Finish with a single line formatted EXACTLY as:
       FINAL ANSWER: <answer>
1331   Example final answer: rnbqkbnr/pppppppp/8/8/8/8/PPPPPPPP/RNBQKBNR w KQkq - 0 1
```

C.2 *Motifs*

**Scope and answer format.** *Motifs* targets pattern-level tactics that do not require deep search: absolute pins, skewers, forks, batteries, and enumerating discovered-check and double-check moves. Prompts permit free-form reasoning but require a single terminal line:



`FINAL ANSWER: <answer>`



Gold labels come from deterministic detectors (ray scans and legal-move simulation) and are serialized as canonical strings; multi-item answers are comma-separated.

**Subtask specifications.**

- **Pins (absolute)** — Return every triplet pinning>pinned>target using *squares*; only absolute pins (target is the king) are included.
- **Skewers** — Every attacker>front>back where the front piece is *more valuable* than the back piece (values: P=1, N=B=3, R=5, Q=9, K=100).
- **Forks** — For each forking square, list attacked enemy pieces as forking>sq1-sq2(-sq3...). Inside each fork, target squares are sorted a→h, then 1→8.

- **Batteries** — For each aligned same-color slider group on one line with empty squares between, return sq>sq(>sq...); multiple batteries comma-separated. (Emission follows a deterministic board-scan order.)

- **Discovered checks** — List all UCI moves by the side to move that uncover a rook/bishop/queen check on the enemy king.

- **Double checks** — List all UCI moves that produce two simultaneous checkers after the move.

(Detectors for each subtask are implemented on top of python-chess with explicit line/ray scans and legal-move simulation (Fiekas, 2024b). )

**Generation pipeline.** We stream shuffled Lichess puzzles (FEN,PuzzleId), emit at most one motif item per puzzle ID, and continue until each of the six subtasks reaches its target count (default 100). For each board we run the corresponding detector(s); when a detector yields at least one motif (or checking move), we serialize the answer in the task's canonical format and stop for that puzzle ID. Full algorithm showed by Algo. 2.

---

**Algorithm 2** *Motifs* Tasks Generation

---

1: **Inputs:** Lichess puzzle CSV (FEN, PuzzleId) **Params:** $N_{sample}$=100, seed= 42
2: SEEDEVERYTHING(42); $D \leftarrow$ READPUZZLES()
3: $U \leftarrow \emptyset$  ▷ used puzzle IDs  counts[pin,fork,battery,skewer,discovered_check,double_check] $\leftarrow 0$
4: **for** row $\in D$ **do**
5:   **if** row.PuzzleId $\in U$ **then continue**
6:   **end if**
7:   $B \leftarrow$ BOARD($row$.FEN)
8:   **for** g $\in$ {PIN, FORK, BATTERY, SKEWER, DISCOVEREDCHECK, DOUBLECHECK} **do**
9:     **if** counts[g] $\geq N_{sample}$ **then continue**
10:    **end if**
11:    $t \leftarrow$ GENERATE_$g(B, \dots)$                    ▷ returns NONE if no motif
12:    **if** $t \neq$ NONE **then**
13:      append $t$; counts[g]++; $U \leftarrow U \cup$ {row.PuzzleId}; **break**
14:    **end if**
15:  **end for**
16:  **if** all counts $\geq N_{sample}$ **then break**
17:  **end if**
18: **end for**
19: SAVETASKS(motif.jsonl)

---

**Detector definitions.** **Pins.** From every enemy rook/bishop/queen, scan rays; if the first two pieces on a ray are ours and the second is our king, record pinning>pinned>target.

**Skewers.** For each friendly slider, trace an attack ray: if the first enemy piece (*front*) is strictly more valuable than a second enemy piece (*back*) behind it on the same line with no interveners, record attacker>front>back.

**Forks.** Any piece that simultaneously attacks $\geq$ 2 enemy pieces yields one fork: forking>sq1-sq2(-sq3...) with target squares sorted.

**Batteries.** Enumerate ranks, files, and diagonals; whenever $\geq$ 2 same-color sliders appear on a line with empty squares between them (and line-consistent movement), emit the ordered group; deduplicate groups.

**Discovered checks.** Simulate each legal move; if the resulting position is check and some checker is a slider *other than* the moved piece whose line to the enemy king was previously blocked only by the moved piece, record the UCI move.

**Double checks.** Simulate each legal move; if the resulting position has at least two distinct checkers, record the UCI move.

**Canonicalization and scoring.**

- **Answer line:** exact match on `FINAL ANSWER: <answer>`.
- **Separators:** use `>` within a structure (pin/skewer/battery chain), `-` within a fork's target list, and comma+space between multiple motifs/moves.
- **Ordering:** fork targets explicitly sorted (file a→h, then rank 1→8); discovered/double checks output as UCI; other groups follow the detector's deterministic board-scan order. (Typical LLM errors we observed include extra/omitted items and format slips. )

**Prompts and exemplars.** We show prompts of the tasks receptively.

**Pins**

```
You are given a chess position in FEN: <FEN>.
Identify all absolute pins in this position. An absolute pin occurs when a piece
↪ cannot move because it would expose its own king to check.
For each pin, provide the key squares in the format:
↪ pinning_piece>pinned_piece>target_piece (e.g., d1>d7>d8).
Analyze step by step and explain your reasoning.
Finish with a single line formatted EXACTLY as:
FINAL ANSWER: <answer>
If more than one, separate with a comma and a space.
Example final answer: d1>d7>d8, a2>e2>h2
```

**Skewers**

```
You are given a chess position in FEN: <FEN>.
Identify all skewers in this position. A skewer occurs when a more valuable
↪ piece is attacked first and forced to move, exposing a less valuable piece
↪ behind it to be captured.
For each skewer, provide the key squares in the format:
↪ skewering_piece>front_piece>back_piece (e.g., a5>e5>h5).
Analyze step by step and explain your reasoning.
Finish with a single line formatted EXACTLY as:
FINAL ANSWER: <answer>
If more than one, separate with a comma and a space.
Example final answer: a5>e5>h5, h1>h4>h7
```

**Forks**

```
You are given a chess position in FEN: <FEN>.
Identify all forks in this position. A fork occurs when one piece attacks two or
↪ more enemy pieces simultaneously.
For each fork, provide the key squares in the format:
↪ forking_piece>attacked_piece1-attacked_piece2(-attacked_piece3 ...) (e.g.,
↪ e5>e7-f6).
Order attacked pieces alphabetically (a>h, then 1>8).
Analyze step by step and explain your reasoning.
Finish with a single line formatted EXACTLY as:
FINAL ANSWER: <answer>
If more than one, separate with a comma and a space.
Example final answer: e5>e7-f6, f3>e1-g1
```

**Batteries**

```
You are given a chess position in FEN: <FEN>.
Identify every battery (2 or more aligned long-range pieces, e.g., RR/RQ/QQ on
↪ files or ranks; BB/BQ/QQ on diagonals; same color; no pieces between).
Report each battery as the squares of the pieces in alphabetical order (a>h,
↪ 1>8), using '>' to separate squares in a battery and ',' to separate
↪ multiple batteries (e.g., h1>h4).
Analyze step by step and explain your reasoning.
```

```
Finish with a single line formatted EXACTLY as:
FINAL ANSWER: <answer>
If more than one, separate with a comma and a space.
Example final answer: b2>e5>h8, h1>h7
```

**Discovered checks**

```
You are given a chess position in FEN: <FEN>.
Identify all discovered-check moves (your move uncovers a check from a
↪  rook/bishop/queen on the enemy king).
Report each as UCI move (e.g., e2e4).
Analyze step by step and explain your reasoning.
Finish with a single line formatted EXACTLY as:
FINAL ANSWER: <answer>
If more than one, separate with a comma and a space.
Example final answer: e2e4, c2b1q
```

**Double checks**

```
You are given a chess position in FEN: <FEN>.
Identify all moves that deliver a double check (two pieces give check after the
↪  move).
Report each as UCI move (e.g., g1f3).
Analyze step by step and explain your reasoning.
Finish with a single line formatted EXACTLY as:
FINAL ANSWER: <answer>
If more than one, separate with a comma and a space.
Example final answer: g1f3, a2a1q
```

C.3 *Short Tactics*

**Scope and answer format.** *Short Tactics* consists of short, single–move tactics drawn from Lichess puzzles. Each item asks for the *best move* for the side to play, output as a single UCI move on the terminal line:

```
FINAL ANSWER: <answer>
```

All prompts are constructed and ship with UCI exemplars. Gold answers are taken directly from the puzzle principal variation (PV) after a preparatory "pre-move" step (as noted by Lichess puzzle database on how to preprocess data, see https://database.lichess.org/#puzzles).

**Subtask specifications.** We provide two families of subtasks:

- **Rating-split**: four levels determined by the puzzle rating — beginner ($\leq$ 999), intermediate ($\leq$ 1499), advanced ($\leq$ 1999), and expert ($\geq$ 2000).

- **Theme-split**: per-theme best-move tasks. Themes including: fork, exposedKing, attraction, discoveredAttack, sacrifice, defensiveMove, intermezzo, pin, mateIn1, smotheredMate, zugzwang, mateIn2, capturingDefender, backRankMate, xRayAttack, skewer, hangingPiece, mateIn3, advancedPawn, queensideAttack, trappedPiece, promotion, deflection and doubleCheck, in total 20 themes.

**Generation pipeline.** A puzzle row is eligible only if it passed rate limit filter, popularity filter and its PV length is under our threshold. Detailed algorithm is showed by Algo 3.

---

**Algorithm 3** *Short Tactics* generation

---

1: **Inputs:** puzzle CSV; theme list JSON     **Params:** thresholds for filters
2: SEEDEVERYTHING(seed); $D \leftarrow$ READPUZZLES()
3: $U \leftarrow \emptyset$ *(used PuzzleIds)*
4: **// Rating split**
5: **for** row in $D$ (shuffled) **do**
6:     **if** FAILFILTERS(row) or row.PuzzleId $\in U$ **then continue**
7:     **end if**
8:     $(fen, move) \leftarrow$ MAKEPREMOVE($row$)
9:     $\ell \leftarrow$ RATINGTOLEVEL($row$.Rating)
10:     **if** count$[\ell] < N_{\text{sample\_rating}}$ **then**
11:         emit TASK(tactic_rating_$\ell$, fen, move, row.*); $U \leftarrow U \cup \{row.PuzzleId\}$
12:     **end if**
13:     **if** all levels filled **then break**
14:     **end if**
15: **end for**
16: **// Theme split**
17: $T \leftarrow$ set of allowed themes (JSON)
18: **for** row in $D$ (continue) **do**
19:     **if** FAILFILTERS(row) or row.PuzzleId $\in U$ **then continue**
20:     **end if**
21:     $\mathcal{I} \leftarrow$ themes(row) $\cap T$;
22:     **if** $\mathcal{I} = \emptyset$ **then continue**
23:     **end if**
24:     **for** $t \in \mathcal{I}$ **do**
25:         **if** count$[t] < N_{\text{sample\_theme}}$ **then**
26:             $(fen, move) \leftarrow$ MAKEPREMOVE($row$)
27:             emit TASK(tactic_theme_$t$, fen, move, row.*, primary_theme=$t$); $U \leftarrow U \cup \{row.PuzzleId\}$
28:             **break**
29:         **end if**
30:     **end for**
31:     **if** all themes filled **then break**
32:     **end if**
33: **end for**
34: SAVETASKS(tactic.jsonl)

---

**Canonicalization and scoring.**

- **Answer type**: single — one UCI move exactly (include promotion suffix, e.g., d7d8q, if applicable).

- **Evaluation**: exact string match on the terminal line after FINAL ANSWER:.

- **Formatting**: no extra tokens, punctuation, or SAN; UCI only. Prompts include exemplars to reduce formatting errors.

**Prompts and exemplars.**    We show prompts of two families respectively.

**Best move (rating-split)**

```
You are given a chess position in FEN: <FEN>.
Find the best move for the side to play.
Analyze step by step and explain your reasoning.
Finish with a single line formatted EXACTLY as:
FINAL ANSWER: <answer>
Use UCI notation (e.g., e2e4, c2b1q) for the final answer.
```

**Best move (theme-split)**

```
You are given a chess position in FEN: <FEN>.
Find the best move for the side to play.
Analyze step by step and explain your reasoning.
Finish with a single line formatted EXACTLY as:
```

```
FINAL ANSWER: <answer>
Use UCI notation (e.g., e2e4, c2b1q) for the final answer.
```

### C.4  *Position Judgment*

**Scope and answer format.**  Class 4 asks models to judge a static position's strength in *centipawns from White's perspective*. For each item we present a FEN and a fixed 5-choice option set; the model must output exactly one number (as text) on the terminal line FINAL ANSWER: <answer>. Gold labels are derived from engine evaluations and mapped to the nearest option. This category's items are all single answer questions.

**Categories and option set.**  We bucket positions by their centipawn (cp) score into five disjoint ranges (White's perspective), and we always show the same choice list $\{-400, -200, 0, 200, 400\}$.

| Category | cp range | Task type |
|---|---|---|
| losing | $[-450, -350]$ | losing |
| disadvantage | $[-250, -150]$ | disadvantage |
| neutral | $[-50, 50]$ | neutral |
| advantage | $[150, 250]$ | advantage |
| winning | $[350, 450]$ | winning |

At label time we snap the engine's cp to the *closest* option; ties break toward the first option in the list (e.g., $+100 \rightarrow 0$, $-100 \rightarrow -200$).

**Gold construction (engine stream $\rightarrow$ tasks).**  We process the evaluation stream from Lichess eval database (https://database.lichess.org/#evals) and parse each line to extract: the FEN, the *deepest* entry in evals (by depth), and its first principal variation pvs[0] (skipping lines that indicate mate). From that PV we take the centipawn score (cp), the PV line (line), and metadata such as depth and knodes. We then compute a position hash from the first four FEN fields (piece placement, side to move, castling, en passant) to deduplicate and bucket the position into one of the five categories above. For each category we sample up to tasks_per_category positions and create one task per position.

**Canonicalization & scoring.**  Prompts allow free-form reasoning, but the grader expects a *single* numeral chosen from the option set on the terminal line. The correct answer is the nearest option to the engine cp; we attach the exact engine cp and PV depth in the item metadata.

**Prompt.**

```
You are analyzing a chess position in FEN: <FEN>.
Estimate the Stockfish evaluation in centipawns (from White's perspective).
↪  Think deeper about this position: Don't just evaluate the current board
↪  state. Consider what the most likely moves are for both sides and how the
↪  centipawn evaluation would change as the position develops. Analyze a moves
↪  ahead - what does the future of this position look like? How would a strong
↪  engine assess this position after calculating many moves deep?
Analyze step by step and explain your reasoning.
Finish with a single line formatted EXACTLY as:
FINAL ANSWER: <answer>
Choose the closest evaluation from the following options: -400, -200, 0, 200,
↪  400.
Example final answers: 400
```

**Algorithm.**  We have two steps, first select position, then create the problem. Detailed algorithm is showed below.

---

**Algorithm 4** *Position Judgment* — Position Selection

---

1: **Input:** stream of engine evaluations on positions; target count $T$ per category
2: **Categories:** five disjoint cp ranges (White's view): `losing`, `disadvantage`, `neutral`, `advantage`, `winning` (see main text)
3: **Output:** balanced set $\mathcal{S}$ of positions with category labels
4: Initialize empty buckets $\mathcal{B}[c]$ for each category $c$; initialize empty set of seen position IDs
5: **for each** evaluation record **do**
6:     Extract the deepest available engine evaluation (centipawns $cp$ and principal line); skip mate scores
7:     Normalize the position ID from the FEN's piece/turn/castling/en-passant fields; **continue** if already seen
8:     $c \leftarrow$ CATEGORYFROMCP$(cp)$; **continue** if $c$ is undefined
9:     **if** $|\mathcal{B}[c]| < T$ **then**
10:         append record to $\mathcal{B}[c]$; mark position ID as seen
11:     **end if**
12:     **if** all categories have $|\mathcal{B}[c]| = T$ **then break**
13:     **end if**
14: **end for**
15: **return** $\mathcal{S} \leftarrow \bigcup_c \mathcal{B}[c]$

---

**Algorithm 5** *Position Judgment* — Item Assembly

---

1: **Input:** balanced set $\mathcal{S}$ of labeled positions
2: **Constant option set:** $\mathcal{O} = \{-400, -200, 0, 200, 400\}$
3: **Output:** multiple-choice judgement items with exact-match answers
4: **for each** position $x \in \mathcal{S}$ with engine $cp(x)$ and category $c(x)$ **do**
5:     $a(x) \leftarrow \arg\min_{o \in \mathcal{O}} |cp(x) - o|$           ▷ nearest option snap; ties break by fixed option order
6:     Build the prompt with the shared prompt constructor, appending the option list $\mathcal{O}$ and the terminal line format `FINAL ANSWER: <answer>`
7:     Store one item with fields: *type*=judgement_$c(x)$, *input*=FEN, *options*=$\mathcal{O}$, *answer*=$a(x)$, and metadata (cp, depth, principal line)
8: **end for**

---

### C.5 *Semantic*

**Scope and task family.** *Semantic* evaluates *semantic understanding* by asking a model to pick, from several alternatives, the natural-language commentary that best describes a concrete chess *position and the just-played move*. Each item shows: (i) a FEN snapshot *before* the move, (ii) the move in UCI, and (iii) 4 textual options (one gold, three distractors). The model must output a single letter A–D on the terminal line `FINAL ANSWER: <letter>`. Items are constructed from real human comments extracted from PGN mainlines and then curated via a cleaning and relevance-judgment pipeline.

**Pipeline overview (data → tasks).**

1. **Extract & keyword-gate comments** from PGN mainlines. For each commented move we keep: raw comment, `move_uci`, FENs before/after, SAN history up to the move, move number, side to move, moved piece, players, and lightweight *thematic keywords* (e.g., *fork*, *outpost*, *zugzwang*). We only keep comments with $\geq 5$ words that match at least one chess keyword (default keyword list provided).

2. **Clean comments** with an offline LLM (`Qwen/Qwen3-30B-A3B-Instruct-2507`, inferenced by vllm (Yang et al., 2025; Kwon et al., 2023)) pass through: replace mentions of the two players with "White/Black"; if any *other* person name appears, SKIP; normalize figurines/markup; optionally convert first-person pronouns when the annotator equals a player. Output is either the cleaned text or SKIP.

3. **Judge relevance** with a second LLM (`Qwen/Qwen3-30B-A3B-Instruct-2507`, inferenced by vllm (Yang et al., 2025; Kwon et al., 2023)) pass (KEEP/DROP), preceded by a chess-aware heuristic that quickly drops generic or non-chess content (regex for SAN/UCI, chess terms, square names). Only KEEP items continue.

4. **Assemble MCQ tasks** for four *distractor strategies*: `easy_random`, `keyword`, `piece_stage`, `embedding`. We compute dense embeddings once (cached) to support semantic neighbors; options are shuffled and mapped to a letter label with exact-match grading.

**Prompt.**

```
You are given a chess position in FEN: <FEN_before>
A player makes the move: <move_uci>

Select the commentary that best describes this position and move.

Options:
A. <option A>
B. <option B>
C. <option C>
D. <option D>

Analyze step by step and explain your reasoning.
Finish with a single line formatted EXACTLY as:
FINAL ANSWER: <letter>
```

**Distractor strategies.**

- `easy_random`: sample distractors uniformly from other cleaned+kept comments. Backfilled if collisions reduce uniqueness.
- `keyword`: pick comments sharing at least one thematic keyword with the gold; random fill if needed.
- `piece_stage`: match by (moved piece type, game phase bucket = opening/middlegame/endgame from move number) to increase surface plausibility; then fall back to same-piece or random.
- `embedding`: nearest neighbors by cosine similarity over sentence embeddings.

All strategies enforce *unique* options and ensure the gold string appears exactly once.

**Configuration.** We use `N_sample_mcq = 100` per variant, `num_options = 4`, `num_distractors = 3`, random sampling enabled, phase thresholds `opening=12`, `middlegame=30`, and embeddings via `Qwen/Qwen3-Embedding-8B` (Zhang et al., 2025) with normalized cosine retrieval (batched, cached).

**Algorithm. Stage A — Comment extraction & keyword gating.** (PGN → candidates) (Con-

---

**Algorithm 6** ExtractCandidates

---

1: **Input:** PGN mainline games; chess keyword list; min word count $m$ (default $m=5$)
2: **Output:** candidate set $\mathcal{C}$ of {comment, FEN_before/after, move_uci, SAN_so_far, move_no, side, moved_piece, players, keywords}
3: **for each** game $G$ **do**
4:     **for each** mainline move node $u$ with attached text $t$ **do**
5:         compute KEYWORDS$(t)$ using normalized matching; **continue** if none
6:         **keep** if WORDCOUNT$(t) \geq m$; record metadata from board state
7:     **end for**
8: **end for**
9: **return** $\mathcal{C}$

---

structs per-move records, identifies moved piece, and formats PGN-until-move; keywords include tactics/strategy/endgame terms.)

**Stage C — Relevance judgment.** (drop generic/irrelevant) (Combines fast heuristics and strict LLM labeling to retain comments that tie *directly* to the concrete move/position.)

**Stage D — MCQ assembly with distractors.**

---

**Algorithm 7** JudgeRelevance

1: **Input:** $\mathcal{C}'$
2: $\mathcal{R} \leftarrow \emptyset$
3: Heuristic prefilter HEURISTICISRELEVANT(r): require chess tokens (SAN/UCI/squares or chess words);
   drop very short/generic text
4: LLM judge with context (FENs, move, PGN prefix); strict KEEP/DROP
5: **for** $r \in \mathcal{C}'$ **do**
6:    **if not** HEURISTICISRELEVANT(r) **then continue**
7:    **end if**
8:    $\ell \leftarrow$ LLMJUDGE$(r)$
9:    **if** $\ell =$ KEEP **then**
10:        add $r$ to $\mathcal{R}$
11:    **end if**
12: **end for**
13: **return** $\mathcal{R}$

---

**Algorithm 8** BuildMCQ

1: **Input:** relevant comments $\mathcal{R}$; number of options $n=4$; distractors $d=3$; variants $\mathcal{V}$
2: $\mathcal{T} \leftarrow \emptyset$
3: Compute indices by keyword/piece/(piece, stage); compute or load embedding matrix (unit-normalized)
   for semantic neighbors
4: **for** each base item $i \in \mathcal{R}$ and variant $v \in \mathcal{V}$ **do**
5:    select $d$ distractors via $v \in \{$random, keyword, piece_stage, embedding$\}$ with fallbacks
6:    ensure uniqueness and pad from the pool if needed; shuffle options; map gold $\rightarrow \{A, \ldots, D\}$
7:    emit one task with prompt (FEN_before, move_uci, option list), letter key, and metadata; add to $\mathcal{T}$
8: **end for**
9: **return** $\mathcal{T}$

---

# D  REPRESENTATIVE ITEMS

We show one fully formatted example from each *ChessQA* category. Each example includes the public FEN/move context and the gold exact-match final line used by the grader. Prompts follow the paper's "single terminal line" convention FINAL ANSWER: <...>.

♟ **Structural**

**Task ID:** `structural_state_tracking_mid_xxxx`

**Prompt**

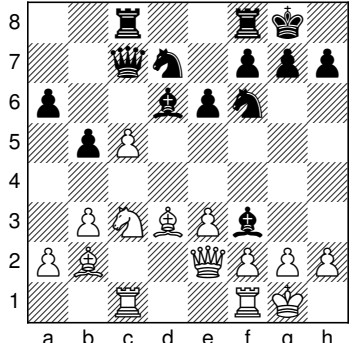

*Diagram for visualization only; the figure is not part of the benchmark prompt or evaluation.*

```
Given an initial FEN and a sequence of UCI moves, apply the moves
    in order and output the exact resulting FEN.
Initial FEN: 2r2rk1/2qn1ppp/p2bpn2/1pP5/8/1PNBPb2/PB2QPPP/2R2RK1
    w - - 0 16
Moves (UCI): e2f3 d6h2 g1h1 h2e5 b3b4 f8d8 d3b1 c7c6 f3h3 d7f8
Analyze step by step and explain your reasoning.
Finish with a single line formatted EXACTLY as:
FINAL ANSWER: <answer>
Example final answer: rnbqkbnr/pppppppp/8/8/8/8/PPPPPPPP/RNBQKBNR
    w KQkq - 0 1
```

**Ground Truth**

```
FINAL ANSWER: 2rr1nk1/5ppp/p1q1pn2/1pP1b3/1P6/2N1P2Q/PB3PP1/1
    BR2R1K w - - 5 21
```

♘ **Motifs**

**Task ID:** motifs_pin_xxxx

**Prompt**

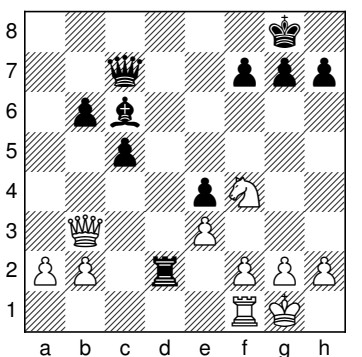

*Diagram for visualization only; the figure is not part of the benchmark prompt or evaluation.*

```
You are given a chess position in FEN: 6k1/2q2ppp/1pb5/2p5/4pN2/1
    Q2P3/PP1r1PPP/5RK1 w - - 0 23. For each pin, provide the key
    squares in the format: pinning_piece>pinned_piece>
    target_piece (e.g., d1>d7>d8).
Analyze step by step and explain your reasoning.
Finish with a single line formatted EXACTLY as:
FINAL ANSWER: <answer>
If more than one, separate with a comma and a space.
```

**Ground Truth**

```
FINAL ANSWER: b3>f7>g8
```

## ♖ Short Tactics

**Task ID:** short_tactics_theme_mateIn2_xxxx

### Prompt

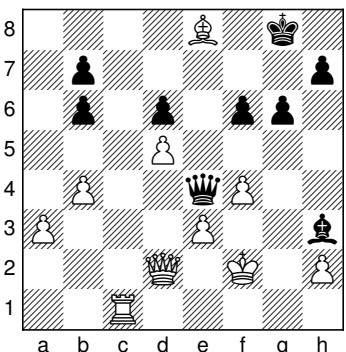

*Diagram for visualization only; the figure is not part of the benchmark prompt or evaluation.*

```
You are given a chess position in FEN: 4B1k1/1p5p/1p1p1pp1/3P4/1
    P2qP2/P3P2b/3Q1K1P/2R5 b - - 0 29.
Find the best move for the side to play.
Analyze step by step and explain your reasoning.
Finish with a single line formatted EXACTLY as:
FINAL ANSWER: <answer>
Use UCI notation (e.g., e2e4, c2b1q) for the final answer.
```

### Ground Truth

```
FINAL ANSWER: e4g2
```

## ⚖️ Position Judgment

**Task ID:** position_judgement_advantage_xxxx

### Prompt

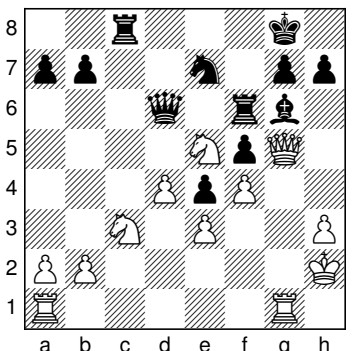

*Diagram for visualization only; the figure is not part of the benchmark prompt or evaluation.*

```
You are analyzing a chess position in FEN: 2r3k1/pp2n1pp/3q1rb1/4
    NpQ1/3PpP2/2N1P2P/PP5K/R5R1 w - -.
Estimate the Stockfish evaluation in centipawns (from White's
    perspective). Think deeper about this position: Don't just
    evaluate the current board state. Consider what the most
    likely moves are for both sides and how the centipawn
    evaluation would change as the position develops. Analyze a
    moves ahead - what does the future of this position look like
    ? How would a strong engine assess this position after
    calculating many moves deep?
Analyze step by step and explain your reasoning.
Finish with a single line formatted EXACTLY as:
FINAL ANSWER: <answer>
Choose the closest evaluation from the following options: -400,
    -200, 0, 200, 400.
Example final answers: 400
```

### Ground Truth

```
FINAL ANSWER: 200
```

**💬 Semantic**

**Task ID:** semantic_easy_random_xxxx

**Prompt**

*Diagram for visualization only; the figure is not part of the benchmark prompt or evaluation.*

```
You are given a chess position in FEN: r2q1rk1/pp3ppp/2p1bnn1/2
    P1p3/8/1BP2pNP/PP3PP1/R1BQ1RK1 w - - 0 14
A player makes the move: d1f3
Select the commentary that best describes this position and move.
Options:
A. When White touched the knight, White took his pen and intended
    to write down 30 Ne7. After White put the knight on d8, he
    kept looking at the board with his pen hanging in the air.
    This was indeed an unpleasant move to face in his slight time
    trouble.
B. Now the bishop that White failed to capture causes White a lot
    of trouble. As the following variations show, Black has a
    very strong attack no matter what White does.
C. After the skirmish in the centre, the white queen is brought
    into the struggle with effect.
D. Black has a lot of space on the K-side and he can start
    counterplay by g6-g5 and f5-f4.
Analyze step by step and explain your reasoning.
Finish with a single line formatted EXACTLY as:
FINAL ANSWER: <answer>
Example format: A
```

**Ground Truth**

```
FINAL ANSWER: C
```

# E   EXAMPLE MODEL RESPONSES

We choose the best performed model GPT-5 * to display some example failures that models would have when doing problems.

E.1    WRONG PIECE / SQUARE RECOGNITION

**Category / Task type:** *Motifs* — Task type: skewer
**Error type:** Wrong Answer
**Expected:** f7>g6>h5
**Model:** f7>f6>f4

**Question.**

You are given a chess position in FEN: 6k15qp1p2bp1Np1p1b3P5P2P3p1P11P5K2Q5 b - - 1 42. Identify all skewers in this position. A skewer occurs when a more valuable piece is attacked first and forced to move, exposing a less valuable piece behind it to be captured. For each skewer, provide the key squares in the format: skewering_piece>front_piece>back_piece (e.g., FORMAT_EXAMPLE_PLACEHOLDER). Analyze step by step and explain your reasoning. Finish with a single line formatted EXACTLY as: FINAL ANSWER: <answer> If more than one, separate with a comma and a space.

**Model response.**

Step-by-step reasoning: - Identify line pieces: Black has a queen (f7) and bishops (d6,d5). White has a queen (c1). No rooks or bishops for White. - A skewer requires a line piece to attack a more valuable enemy piece that has a less valuable enemy piece directly behind it on the same line (with only empty squares in between). - Check Black queen f7: - Along the f-file downward: f6 (White knight) is attacked; behind it on the same file is f4 (White pawn). Knight (3) > Pawn (1), and the line is clear (f5 empty). This is a valid skewer: f7>f6>f4. - Other directions from f7 are blocked or do not produce a more-then-less value alignment. - Check Black bishops d6, d5: Their attacks hit either single pieces or alignments with equal/greater value behind (e.g., d6→f4→g3 are pawns; not a skewer), or are blocked. - Check White queen c1: Its rays are blocked (e3 pawn or b2 pawn) or hit no suitable pair of Black pieces. No skewer from White. Only one skewer exists. FINAL ANSWER: f7>f6>f4

E.2    *Short Tactics* LEGALITY MISTAKES

**Category / Task type:** *Short Tactics* — Task type: expert
**Error type:** Wrong Answer
**Expected:** d7f6
**Model:** g7g5

**Question.**

You are given a chess position in FEN: 3r1q1k/p2nNpp1/1p2p2p/8/nPP4Q/P4N2/5PPP/3R2K1 b - - 3 24. Find the best move for the side to play. Analyze step by step and explain your reasoning. Finish with a single line formatted EXACTLY as: FINAL ANSWER: <answer> Use UCI notation (e.g., e2e4, c2b1q) for the final answer.

**Model response.**

Step-by-step reasoning:- Reconstructing the position: Black pieces include Kg8-h8, Qf8, Rd8, knights at a4 and d7, pawns a7, b6, e6, f7, g7, h6. White has Kg1, Qh4, Rd1, knights e7 and f3, pawns a3, b4, c4, f2, g2, h2.- Critical tactical motif: If Black moves the d7-knight (e.g., ...Ne5 or ...Nf6), White can reply Rxd8 Qxd8 and then Ng6+ followed by Qxd8, winning Black's queen. So moving the d7-knight is dangerous.- The e7-knight is guarded only by the queen on h4 along the diagonal h4–g5–f6–e7. If Black attacks the queen with ...g5, White must either: - Move the queen off that diagonal (e.g., Qg3/Qh5/Qf4), when Black plays ...Qxe7 winning the knight; or - Capture en passant-like with 25 Nxg5, but then ...Qxe7 again wins the e7-knight because the diagonal is blocked by the knight on g5. White may grab a pawn or two with Qxh6+ afterward, but Black remains up material.- Alternative attractive idea 24...Nc3 threatens Nxd1, and 25 Rc1? allows ...Ne2+ and ...Nxc1 winning the exchange, but White can avoid this with 25 Re1/Rf1, keeping things more balanced. Therefore 24...g5 is more forcing and clearly wins material.- Bad move refutation example:

24...Ne5? loses to 25 Rxd8 Qxd8 26 Ng6+ followed by Qxd8, winning Black's queen. Best move:- 24...g5! attacks the queen, disconnects it from supporting e7, and wins the e7-knight next move in all lines. FINAL ANSWER: g7g5

### E.3 Sound analysis, wrong final answer

**Category / Task type:** *Short Tactics* — Task type: theme queensideAttack
**Error type:** Wrong Answer
**Expected:** d7f6
**Model:** g7g5

**Question.**

You are given a chess position in FEN: 6k14Rpp1p6pq3p31rp1P2PpPQ2PP1P71K5R b - - 0 30. Find the best move for the side to play. Analyze step by step and explain your reasoning. Finish with a single line formatted EXACTLY as: FINAL ANSWER: <answer> Use UCI notation (e.g., e2e4, c2b1q) for the final answer.

**Model response.**

Let's decode the position. - White king: b1; queen: c3; rooks: e7 and h1; pawns: a2, b3, e4, f3, g3, h4. - Black king: g8; queen: a5; rooks: b4 and c4; pawns: a6, a3, d4, e5, f7, g7, h6. - Black to move. Key tactical feature: White's queen on c3 sits directly in front of Black's rook on c4. The move Rxc3 simply wins White's queen outright. There are no interferences or pins: the path from c4 to c3 is clear, and b4 continues to block the a5–c3 diagonal, so no self-pins or discovered checks arise. After 30...Rxc3, White cannot recapture the rook with the queen (it's gone), and any compensation ideas (like Re8+ or Rxf7) are completely insufficient; Black remains up a full queen with a safe king. Alternative tempting idea: 30...Rxb3+ 31.axb3 Qxc3 also wins the queen, but at the cost of sacrificing a rook first, leaving Black up a queen for a rook (smaller material gain) and giving White extra activity. Therefore, directly 30...Rxc3 is superior. Conclusion: The cleanest and strongest move is capturing the queen immediately. FINAL ANSWER: c4c3

### E.4 Saying "no answer" when an answer exists

**Category / Task type:** *Motifs* — Task type: skewer
**Error type:** Wrong Answer
**Expected:** d7f6
**Model:** None

**Question.**

You are given a chess position in FEN: r4rk15pppp1p5q1p1p3N1P1P1n11P6P1Q2PPP3R1RK1 w - - 0 17. Identify all skewers in this position. A skewer occurs when a more valuable piece is attacked first and forced to move, exposing a less valuable piece behind it to be captured. For each skewer, provide the key squares in the format: skewering_piece>front_piece>back_piece (e.g., d1>d7>d8, a2>e2>h2). Analyze step by step and explain your reasoning. Finish with a single line formatted EXACTLY as: FINAL ANSWER: <answer> If more than one, separate with a comma and a space.

**Model response.**

Step-by-step reasoning: - A skewer needs three things in a straight line: an attacking line-piece (rook/queen/bishop), a more valuable enemy piece in front, and a less valuable enemy piece behind it, with no blocking pieces between the two enemy pieces along that line. - Check black king alignment: g8 (king) with f8 (rook) behind along the 8th rank would be a classic K>R skewer if White could attack from h8 along the rank. However, White has no way to place a rook/queen on the 8th rank from the h-file or otherwise open that line now; the route is blocked and unreachable. - Check a-file: a5 (black queen) is in front of a6 (black pawn), but White cannot line up a rook/queen on the a-file from below because a2 (white pawn) and a4 (white knight) block any rook/queen from the a-file; thus no executable skewer. - Other ranks/files/diagonals: No other pairs of black pieces are aligned with a

higher-value piece in front of a lower-value one such that White's rooks/queen can attack along the required line. Black similarly has no current skewer against White (no line-piece is attacking a more valuable white piece with a less valuable white piece behind it). - Conclusion: There are no actual skewers present or immediately playable in this position. FINAL ANSWER: none

