# OpenReview forum: "ChessQA: Evaluating Large Language Models for Chess Understanding"
_ICLR.cc/2026/Conference — Submitted to ICLR 2026_

### Official Review · Reviewer_6T1y · 2025-10-28

**Soundness:** 3
**Presentation:** 2
**Contribution:** 2
**Rating:** 4
**Confidence:** 3

**Summary:**

The paper introduces ChessQA, a new benchmark for measuring chess understanding and ability. The motivation for a new chess benchmark is that existing works only cover parts of the spectrum of skills or are gameplay evaluations. ChessQA has five levels of tasks: structural, motifs, short tactics, position judgment, and semantic. Evaluation is done either as exact match with the correct answer or through multiple choice. Many of the leading LLMs are evaluated, and scores are roughly consistent with behaviour on general benchmarks: GPT-5 leads the benchmark, while weaker, open-source models struggle.

**Strengths:**

- The benchmark is an improvement over existing knowledge-based chess benchmarks, which only cover specific areas. ChessQA is significantly broader in scope covering five topics in detail. The set-up of the benchmark seems rigerous.
- Since the benchmark is broken into five components, performance can be analysed in detail, and areas of weakness located.
- There is significant interest in tracking LLMs' chess abilities from the wider AI community, though this interest is mainly tokenistic due to historical reasoning and the popularity of the game. Performance here is not necessarily the best indicator of performance on reasoning tasks more broadly.

**Weaknesses:**

### General points
- **Motivations:** The paper could benefit from making the necessity of this benchmark clearer. In particular, further discussion and comparison with [1], the head-to-head game evaluation, would be valuable. My thoughts are that gameplay is a better testbed for chess understanding and ability since the metric (game success) implicitly incorporates all of the skills and knowledge we care about measuring. It would be good to see an argument for why this benchmark is relevant in light of this, and how the results of these experiments compare to those from [1]. For example, do the models follow the same order or are there some models which perform well in knowledge-based tasks but poorly in gameplay, etc.
- **Data contamination:** In addition, many of the problems are taken from existing datasets. Were these problems released before the knowledge cut-off in model training? What is the risk that models have been exposed to these specific situations before? I note that this might be a sliding scale since models may have seen the board states in a different context, but not seen the exact question. This seems like another reason why evaluating models in gameplay situations would be better, since you can almost guarantee that a board state is novel. The paper should acknowledge the limitations of reusing existing datasets
- **Future contamination:** What steps have you taken to prevent the datasets from being included in training?
- **Measuring token use:** I am unsure why token use is the most interesting metric to focus on given that some models are smaller but reason for longer while others are larger but do less reasoning. The appendix presents cost analysis, and this seems a more comprehensive metric of resources required to get the result. [This is a relatively minor point compared to the first three].

### Smaller points
- **Statistics:** Reporting standard errors over scores, etc would aid model comparison
- **Examples:** A couple of example questions in the main paper would help the user understand the tasks. The discussion of the 5 levels is fairly long and somewhat repetitive, so could be cut in place of examples.
- **Thinking models:** The paper reports that they consider GPT-5 in a non-thinking state, which is not an option. Perhaps they mean with minimal reasoning? This is still quite different to non-thinking models. The use of the * to denote thinking is okay, but should be explained before it is used.
- **Limitations:** Add a limitations section or discuss in one of the existing sections.

### Typos
- Section 3.1 standardisation of “First” and (i)
- The error analysis section links to the appendix. Is this deliberate?
- L452 has an incorrect quotation mark

[1] Lee et al. 2025:

**Questions:**

-  There is a large interest in chess since it is a strong historical benchmark, and many people play chess. However, the claims that this is an ideal testbed for reasoning more generally perhaps exceed the results. Do the authors intend to present this as a chess-specific benchmark, or are they making broader claims about the comparative reasoning abilities of LLMs? I couldn't quite establish this from the paper. Depending on the answer, the language of the paper could be better calibrated. If they are making broader reasoning claims, then comparisons with other reasoning benchmarks would be appreciated.
- GPT-5 appears to be a significant jump over existing models. Why do the authors think this is? Is this a large jump over previous OpenAI models? If so, do the authors think the benchmark will soon be saturated as frontier labs start doing more RL training? (I think this is just generally interesting rather than a crituqe of the benchmark!)

---

> ### Author Response · Authors · 2025-11-24
> **Response to Weaknesses and Questions**
>
> Thank you for the thoughtful and constructive review. We are glad that you view ChessQA as a rigorous, useful complement to existing chess benchmarks, and we have revised the paper to clarify motivation, address data‑contamination concerns, and improve the statistics and presentation. Below we respond to each of the weaknesses and questions and clarify how the revised manuscript addresses them.
>
> **W1:** Our goal is to complement game‑arena evaluations rather than replace them. Gameplay Elo is an attractive “all‑in‑one” metric, but in current non-agentic text‑only settings, models often start producing illegal or nonsensical moves very early in the game, so Elo collapses many distinct failure modes (parsing the board, respecting legality, evaluating trades, planning, explanation) into a single coarse outcome. ChessQA instead decomposes ability into five interpretable sub‑skills, such as Structural, Motifs, Short Tactics, Position Judgment, and Semantic, that map closely to the skills humans learn as they improve at chess. This decomposition is especially important for our focus on chess understanding and education: a system that reliably explains motifs and evaluations may be more pedagogically valuable than one whose raw Elo is slightly higher but whose reasoning is opaque. We have clarified Section 2 and Table 1 to emphasize that game‑arena benchmarks such as Kaggle Game Arena are a complementary, agentic lens that measures interactive playing strength, whereas ChessQA offers a controlled QA setting with deterministic scoring that isolates where the underlying skills break down.
>
> Your suggestion to compare results more directly with \[2\] is well taken. Qualitatively, we observe that the ordering on ChessQA, large frontier proprietary models ≳ top open‑source models ≫ smaller models, matches public reports on Kaggle Game Arena and on broader reasoning benchmarks, but we have not yet run a formal correlation study. To avoid overstating our claims, we now state explicitly that (a) we currently treat ChessQA as a chess‑specific benchmark; (b) the relationship between ChessQA scores, game‑arena Elo, and general‑reasoning performance is an open empirical question; and (c) a joint analysis across these benchmarks is interesting future work.
>
> **W2 and W3:** Our benchmark prevents data contamination from three aspects. First, we use proprietary source data: part of our source data is from ChessBase 17, which is a licensed and paid database that is unlikely to appear in open web‑scale training corpora \[6\]. Second, we use refreshing source data: part of our source data is from Lichess puzzles and evaluations databases, which are continually updated monthly. This allows us to use “fresh” slices of data that post‑date LLMs’ training cutoff \[4,5\]. Specifically, we use the latest possible positions in Lichess database at the time of constructing ChessQA. We have updated Section 3.2 to reflect how we prevent data contamination.
>
> On the other hand, there are valid concerns that game-based evaluations will be contaminated, where the models can simply generate the memorized next moves from pre-training data, especially in the openings. In other words, the early openings positions are almost certainly contaminated. Moreover, game-based evaluation may be biased towards popular openings and common patterns in their pre-trained data, resulting in limited coverage of chess understanding capabilities. In ChessQA, we transform raw chess games into QA formats with customized functions, ensuring our tested patterns do not exist in the open online sources repeatedly. Even if a model had seen a particular PGN or FEN during pre‑training, it would still have to understand the position and carry out reasoning to answer the exact question. We also ensure comprehensive task and position coverage, which is inherently more suitable for evaluating general chess understanding capabilities in LLMs.
>
> **W4:** Tokens are a natural unit for interpreting test‑time scaling: Section 4.2 and Figure 5 show how enabling reasoning modes substantially increases tokens per problem while also increasing accuracy, which is central to our analysis.
>
> We agree that cost analysis is interesting as well. However, prices differ across providers and can change over time, especially when using a request routing system such as OpenRouter. Therefore, the cost-per-problem figure might not rigorously reflect the cost efficiency. We thus choose to focus on token efficiency and include the token-per-problem figure in the main content, as token usage is vendor‑agnostic. We have added this as a limitation in Section 6\.

---

> ### Author Response · Authors · 2025-11-24
> **Supplement**
>
> **Smaller Point1:**  We clarify in Section 6 that due to limited budgets, we do not run statistical tests for every model, and we explicitly position this as a limitation. We have run more experiments on a representative model according to your suggestion: Appendix B.3 reports stochastic variation for non‑thinking Claude Haiku‑4.5 across three independent runs, providing mean ± standard deviation and min–max ranges per category and overall (e.g., 27.8% ± 0.4 overall). Moreover, Appendix B.2 now presents results with different reasoning budgets using Claude Haiku‑4.5\*. Performance rises monotonically as we raise the reasoning effort from minimal/low to medium and high (28.9%, 49.4%, and 54.0%, respectively); item‑paired t‑tests over 3 independent runs per setting indicate the low-to-medium jump is highly significant and the medium-to-high improvement is smaller but still significant.
>
> **Smaller Point2:**  We agree that concrete examples help readers understand what tasks look like. However, it’s challenging to include examples in the main content. In response, we added a new section named “Representative Items” (Appendix D) that shows one fully formatted example from each category, including the FEN, prompt, context, chessboard image, and the ground truth. We also point to Appendix D explicitly from Section 3 after the introduction to each category, while keeping the detailed task specifications and grading rules in Section 3 and Appendix C.
>
> **Smaller Point3:** We apologize for the confusion. In the revised experimental setup (Section 4.1), we clarify that “GPT‑5” in our tables refers to the non‑thinking “GPT‑5-Chat” endpoint officially named by OpenRouter, while “GPT‑5\*” denotes the same base model with OpenAI’s reasoning mode enabled and a larger thinking‑token budget. We ensure the “\* denotes thinking enabled” convention is introduced in text before it appears in Figures 3–7 and repeat it in the captions.
>
> **Smaller Point4:**  We have added a dedicated Limitations section (Section 6). There we explicitly note that (i) we do not report a single human baseline because of the enormous spread in human chess strength; instead we rely on Lichess puzzle ratings and engine centipawn evaluations as human‑calibrated difficulty anchors (ii) statistical testing is limited to selected models and ablations; (iii) Short Tactics caps puzzle depth and Position Judgment uses relatively coarse centipawn buckets, leaving room for future refinements; (iv) cost-per-problem is provider-specific and thus not rigorous, and (v) the work is diagnostic rather than interventional.
>
> **Typos are fixed, thanks for pointing them out\!**

---

> ### Author Response · Authors · 2025-11-24
> **Supplement**
>
> **Q1:** We intend to present ChessQA as a chess‑specific benchmark, instead of a universal test of general reasoning ability. Admittedly, our claim of chess being an “ideal” testbed to evaluate LLM reasoning appears to be too strong. Accordingly, we have updated the manuscript to weaken our claim. For example, we have modified “Chess provides an ideal testbed for evaluating the reasoning, modeling, and abstraction capabilities of large language models (LLMs)” to “Chess has played an important historical role in AI development” in the abstract as another way to motivate the importance of chess in AI.
>
> However, we respectfully suggest that being chess-specific does not conflict with evaluating the reasoning capabilities of LLMs. For example, math-specific datasets like MATH \[3\] have been widely used for evaluating LLM reasoning abilities. Chess has several desirable properties as a domain for assessing LLM capabilities. It is a neatly circumscribed, perfect-information, and fully objective domain with clear states, actions, goals, and rewards. Despite this, it is incredibly deep and admits a huge variety of skill levels. People are also still very active in playing chess, and a near-infinite supply of human chess data is available online. If we consider the Position Judgement task, even though LLM may already have chess **knowledge**, calculating centipawn advantage requires long-term planning and implicit search, which necessitates **reasoning** beyond simple chess knowledge. Empirically, the ranking we observe on ChessQA (frontier proprietary models ≥ top open‑source models ≫ smaller models) qualitatively matches public reports on general reasoning focused benchmarks such as GPQA \[1\] and MATH \[3\]. We consider cross‑benchmark comparison as an open research direction for future work.
>
> **Q2:** We found it very interesting as well\! As pointed out in \[7\] GPT-3.5-Turbo is much better than GPT-4 in chess. Therefore, our educated guess is that OpenAI has done something specific to chess, but they chose not to do that anymore when updating to GPT-4. It’s possible that they bring the chess-specific optimization back in GPT-5, resulting in much better performance than other models. On the other hand, we note that GPT-5 only scored about 40% on Position Judgment. Therefore, we believe that ChessQA (and chess understanding in general) is far from being saturated and should remain informative as frontier models undergo further chess-specific optimization, such as the mentioned RL training. To further support this, we constructed an additional test set of 100 hard Lichess puzzles with ELO ratings over 3000\. GPT‑5\* with high reasoning effort solved only 26 out of 100 puzzles (averaging \~24k tokens per puzzle\!), as measured by first move correctness. Such results confirm that expert‑level tactics remain challenging even for the best model we tested, and LLM chess understanding competence still needs to be improved.
>
> **References**
>
> \[1\] Hendrycks, Dan, et al. "Measuring mathematical problem solving with the math dataset." arXiv preprint arXiv:2103.03874 (2021).
>
> \[2\] Lee et al. “Chess Text Input” (Kaggle Game Arena benchmark), 2025\.
>
> \[3\] Rein et al. “GPQA: A graduate‑level, google‑proof Q\&A benchmark,” 2024\.
>
> \[4\] Lichess Team. “Lichess Open Database — Evaluations.” 2025\.
>
> \[5\] Lichess Team. “Lichess Open Database — Puzzles.” 2025\.
>
> \[6\] ChessBase GmbH. ChessBase 17: Commenting Functions (Text & Variants). 2023\.
>
> \[7\] Carlini, Nicholas. “Playing Chess with Large Language Models.” nicholas.carlini.com, 22 Sept. 2023, nicholas.carlini.com/writing/2023/chess-llm.html.

---

> > ### Comment · Reviewer_6T1y · 2025-11-27
> >
> > **W1.** Thank you for clarifying the motivations of the benchmark over game‑arena benchmarks. I think it would be good to add this to the paper somewhere as I think those are convincing and it was certainly in my head when reading the paper. Certainly understanding the failure modes of models is a fairly convincing benefit even as model performance increases. I think it would be good to add a more complete comparison between the two settings, even if it is just using computed scores from other research. That analysis feels important when introducing this benchmark, given we have interesting game-arena chess evaluations already.
> >
> > **W2/W3.** On the contamination issue for ChessQA, the fact that ChessBase is proprietary is a fairly weak guarantee that it hasn't been included in pre-training. On the other hand, the fact that some problems are more recent is. Would it be possible to do a study of included problems introduced before and after a training cut-off to demonstrate that there is no statistically significant increase in performance on data possibly included in pre-training?
> >
> > You make a convincing point about data contamination being a problem for game-play benchmarks. I assume this is only a problem if all models are in the weak regime, where they could be easily exploited with these popular openings. Are the best models beyond this, or is this likely for state-of-the-art models?
> >
> > **W4.** Sure, this is a relatively small point and I am happy with this.
> >
> > **SP1.** Do you need to do repeats to measure the standard errors here? You should be able to get estimates without this. If there are multiple problem types or different scoring functions, then perhaps a bootstrap method would work.
> >
> > **SP2.** Thanks for adding this and I understand the examples are fairly complex here.
> >
> > **SP3.** Thanks for clarifying here! I wasn't aware of the non-reasoning variant of GPT-5 stand corrected. I feel like the term ``GPT‑5" generally is used to refer to the default medium reasoning effort when using the GPT‑5 endpoint. I wonder what OpenAI did in their model release? I don't have strong preferences on any changes here but feel most readers would be expecting GPT‑5 to refer to the reasoning model.
> >
> > **SP4.** Thanks. Yes, a general human baseline would make no sense. Thought it might be interesting to see how the models perform against different thresholds of human ability, e.g. like how Olympiad benchmarks often try to compare to different colour medals.
> >
> > **Q1.** Thanks - this is much better calibrated. I completely agree that chess has lots of reasoning components, which makes it an applied reasoning task, but I don't think it is a good test for reasoning as a whole since it is too specific. I was just unsure from the language whether you were presenting it as a reasoning benchmark or a chess-specific benchmark, so this has answered the question.
> >
> > **Q2.** Interesting. Adding the extra hard test set seems like a good addition.

---

> ### Author Response · Authors · 2025-12-03
>
> #### Thank you for your continued engagement and discussion, and we are happy to hear that our rebuttal addressed most of your concerns.
>
> **W1.** Thanks for recognizing our motivation for proposing ChessQA with the existence of game‑arena benchmarks. We have added the discussion to the end of the related work section (highlighted in blue). We present the model rankings by GameArena and ChessQA in the following table (enabled thinking for all models). Due to the rapid development of frontier models, there are only 4 overlapping models in GameArena and ChessQA. We can see a clear correlation between game-play and question-answering evaluation, while we also observe slightly different rankings, which suggests the two benchmarks are focusing on different aspects of chess understanding. We regard cross-benchmark evaluation as an important future work.
>
> | Model | Rank by GameArena | GameArena | Rank by ChessQA | ChessQa |
> | :---- | :---- | :---- | :---- | :---- |
> | GPT-5 | 1 | 1275 (-81/+91) | 1 | 79% |
> | Gemini 2.5 Pro | 2 | 1004 (-76/+88) | 3 | 43% |
> | DeepSeek-R1 | 3 | 398 (-70/+83) | 2 | 52% |
> | Gemini 2.5 Flash | 4 | 0 (-0/+0)  | 4 | 29% |
>
> **W2/W3.** We agree that using more recent data after the training cut-off is a strong guarantee (which we did), while proprietary source data is not. We have conducted an additional statistical analysis to empirically measure model performance on data from before and after the relevant training cut off. As Claude Haiku 4.5’s knowledge cutoff is February 2025 \[2\], we decompose the Short Tactic (900 items) into two separate raw datasets to directly test for potential training-data contamination: puzzles dated before 2025-01-02 \[3\], and puzzles dated from October \[4\] through November 2025 \[5\], for which we applied an additional filter to ensure no earlier entries were included. The model achieved 117/900 on the pre-cutoff puzzles and 113/900 on the post-cutoff puzzles. We evaluated whether this difference was meaningful using a two-proportion z-test (Z \= 0.28, p \= 0.78), a chi-squared test (χ² \= 0.04 p \= 0.83), and Fisher’s exact test (OR \= 1.04, p \= 0.83), none of which indicated a statistically significant difference. Overall, there is **no evidence that model performance differs between puzzles before and after the training cutoff** (p \= 0.78–0.83). These results indicate that the **model did not benefit from any potential exposure to older puzzles during pre-training**, hopefully addressing your concern regarding data contamination.
>
> We would like to clarify that data contamination in the form of exploiting popular openings is an inherent disadvantage of game-play benchmarks, affecting even state-of-the-art models. The improved version of the Kaggle Game Arena \[6\] has identified this problem and instituted a workaround: "To ensure diverse and realistic gameplay, every match begins from one of the 20 most popular positions that can be reached after two plies (one move by White and one move by Black), based on data from a randomly chosen month on Lichess." The fact that this intervention was necessary even for frontier models suggests that current state-of-the-art models still suffer from opening data contamination, rather than this being limited to weaker models. While future models may become more robust to this issue, at present even the best models require such mitigations, highlighting a fundamental limitation of game-play benchmarks that ChessQA avoids through its QA-based evaluation.
>
> **SP1.** A way to measure the standard errors without repeating is to compute the standard errors of binomial proportions. We have added such results in the updated Figure 3 in the revision.
>
> **SP3.** GPT-5 uses “medium” reasoning effort by default, while GPT-5.1 is non-reasoning by default (“none” reasoning effort) \[1\]. We have  added clarification about this in Section 4.1 of the revision (highlighted in blue), and keep the naming strategy in the paper to be consistent with other model names.
>
> \[1\] [https://platform.openai.com/docs/guides/latest-model\#page-top](https://platform.openai.com/docs/guides/latest-model#page-top)
>
> \[2\] [https://www.anthropic.com/transparency](https://www.anthropic.com/transparency)
>
> \[3\] [https://web.archive.org/web/20250113022223/https://database.lichess.org/\#puzzles](https://web.archive.org/web/20250113022223/https://database.lichess.org/#puzzles)
>
> \[4\] [https://web.archive.org/web/20251003004921/https://database.lichess.org/\#puzzles](https://web.archive.org/web/20251003004921/https://database.lichess.org/#puzzles)
>
> \[5\] [https://database.lichess.org/\#puzzles](https://database.lichess.org/#puzzles)
>
> \[6\] [https://www.kaggle.com/benchmarks/kaggle/chess-text-openings/versions/1](https://www.kaggle.com/benchmarks/kaggle/chess-text-openings/versions/1)
>
> \[7\] [https://arxiv.org/abs/2509.02534](https://arxiv.org/abs/2509.02534)

---

### Official Review · Reviewer_mMdF · 2025-10-30

**Soundness:** 3
**Presentation:** 3
**Contribution:** 2
**Rating:** 4
**Confidence:** 3

**Summary:**

The paper introduces ChessQA, a 3,500-item, 50-task benchmark that evaluates LLM chess understanding across five categories (Structural, Motifs, Short Tactics, Position Judgment, and Semantic), intended to span a curriculum from rules to high-level explanation. The authors evaluate 13 contemporary LLMs in text-to-text settings (some with “thinking mode”), report overall low performance with best run 79.3% (thinking enabled GPT-5), analyze performance across categories, study token/cost efficiency, and present qualitative error typologies.

**Strengths:**

* Comprehensive task coverage in chess understanding. ChessQA covers 50 diverse tasks, offering a thorough and fine-grained evaluation of chess-related reasoning capabilities.
* Extensive model evaluation. The benchmark is evaluated across 13 different models, and explicitly accounts for thinking modes, providing a broad perspective on model behavior.
* In-Depth Analysis of Results. The paper conducts detailed analyses, including token efficiency and performance scaling, offering insights into both model capability and cost-effectiveness.

**Weaknesses:**

* Limited per-task coverage. The dataset includes 3,500 examples spanning 50 task types, resulting in an average of only 70 examples per task. This relatively small number may limit reliable evaluation for individual task categories.
* No human baseline. The paper does not report human performance, either from laypeople or domain experts, which makes it difficult to contextualize the difficulty of the tasks and assess how far current models are from human-level understanding.
* Lack of discussion on real-world relevance. The paper does not explore whether chess understanding can transfer to real-world reasoning tasks.
* Position Judgment grid selection lacks justification. The design choices behind the bucket grid used for position judgment tasks do not provide enough support.
* No access to the dataset.

**Questions:**

1. Does strong performance on ChessQA correlate with improved reasoning capabilities in real-world scenarios?
2. What is the rationale behind the choice of bucket grid boundaries for the Position Judgment task? Specifically, why were the thresholds set to {−400, −200, 0, 200, 400}?
3. Does the use of exact-match accuracy, as stated in Section C.1, unfairly penalize models with weaker instruction-following abilities?

---

> ### Author Response · Authors · 2025-11-24
> **Response to Weaknesses and Questions**
>
> We thank the reviewer for the thoughtful review and for highlighting the strengths of ChessQA’s task coverage, model breadth, and analysis. Below we respond to each of the weaknesses and questions and clarify how the revised manuscript addresses them.
>
> **W1:** Our main analyses aggregate results at the category level instead of the subtask level. The number of items in each category is presented in Section 3 and Figure 2: Structural \= 1,100, Motifs \= 600, Short Tactics \= 900, Position Judgment \= 500, and Semantic \= 400\. Among which, most individual subtasks have 100 items; only the 20 theme‑split Short Tactics subtasks have 25 items in order to cover a broad range of chess themes, while the 4 strength‑split Short Tactics subtasks contain 100 items, giving 900 items total in that category. We compute an additional statistical metric to show that our sample sizes can support analysis with category-level aggregation. In particular, the computed binomial standard errors are on the order of 1 to 2.5 percentage points for our sample sizes, much smaller than the 10 to 40‑point gaps between models in Figure 3\. Notably, our benchmark is generated fully automatically, and we have access to virtually unlimited source data, e.g., more than 5 million puzzles and more than 7 billion games on the Lichess Database at the time of composing this response. Although we are able to scale up ChessQA, we have to consider the cost to run our benchmark, both for getting the results presented in this paper and for future results from the community.
>
> **W2:** We anchor difficulty to large‑scale human behavior that is already embedded in our data sources: Lichess puzzle ratings summarize historical attempts of many human players at different strengths, which guides the strength-based separation of subtasks in the Short Tactics category. We agree that an explicit human baseline is a valuable future work, but our budget is limited to conducting such experiments across people with a spread in chess ability. We have included this as a limitation in Section 6\.
>
> **W3 and Q1:** We intend to present ChessQA as a chess‑specific benchmark, instead of a universal test of general reasoning ability. We interpret “real-world scenarios” mentioned by the reviewer as evaluations on other general-purpose reasoning benchmarks. Empirically, the ranking we observe on ChessQA (frontier proprietary models ≥ top open‑source models ≫ smaller models) qualitatively matches public reports on general reasoning focused benchmarks such as GPQA \[6\] and MATH \[7\]. We consider cross‑benchmark comparison as an open research direction for future work.
>
> **W4 and Q2:**  Any multiple‑choice formulation necessarily imposes discrete buckets on a continuous scale, so some choice of thresholds is unavoidable. We adopt the five‑option grid {−400, −200, 0, 200, 400} as a reasonable instantiation out of many. In particular, these values are aligned with common practice in human chess annotation: “clearly worse”(-400), “slightly worse”(-200), “equal”(0), “slightly better”(200), and “clearly better”(400) correspond to standard Numeric Annotation Glyph categories (e.g., “±”, “+/−”) in PGN \[1\], one of the most commonly used chess notation. Modern engines expose centipawn scores that chess practitioners routinely map to these discrete bands when annotating games. For example, Stockfish’s NNUE evaluation \[2\] and the Stockfish 17.1 release notes \[3\] are representative examples of such usage, as is the Lichess Evaluations database \[4,5\], which aggregates engine scores for millions of positions that players mentally convert to corresponding qualitative labels in practice. In Section 3.1, we clarify that our pipeline already supports alternative grids; for example, tightening the buckets to {−200, −100, 0, 100, 200} is described as a built‑in option for raising task difficulty. We have added the abovementioned support for option construction in Section 3.3.
>
> **W5:** The abstract in the initial submission already contains an anonymous link in the sentence “We will release the code, periodically refreshed datasets, and a public leaderboard to support further research.” In the revision, we have made it visually clearer by underlining it in the abstract. The repository includes data files, generation scripts, and evaluation code sufficient to reproduce all reported results.

---

> > ### Author Response · Authors · 2025-11-24
> > **Supplement to "Response to Weaknesses and Questions"**
> >
> > **Q3:** We view basic instruction‑following as part of the competence being evaluated: a chess‑capable model should both reason correctly and follow a simple output format. That said, we took care to ensure that scoring is not dominated by formatting artifacts. As described in Section 4.1 and Appendix C, we always parse the final answer from a clearly marked terminal line and canonicalize it before comparison: multi‑move answers are normalized into sets, piece lists are sorted in a fixed order, and move lists are sorted or generated in deterministic order. Figure 4 breaks down errors into correct answers, wrong but well‑formed answers, format errors, and token‑limit failures; format errors account for only a small share of failures across models, whereas the dominant error types involve board‑state hallucination, illegality, wrong tactical choices, or false “no answer” (further illustrated in Appendix E). In Appendix B.1 we also report robustness checks where we replace the format exemplars in the prompts (ΔExample) and observe that overall accuracy changes are within about one percentage point. Together, these diagnostics suggest that exact‑match scoring is not unfairly punishing instruction‑following; the main performance gaps we observe reflect genuine chess‑reasoning capabilities.
> >
> > **References**
> >
> > \[1\] Edwards, S. J. (1994). Portable Game Notation (PGN) Specification and Implementation Guide.
> >
> > \[2\] Stockfish Developers. “Introducing NNUE Evaluation.” Stockfish Blog, 2020\.
> >
> > \[3\] Stockfish Developers. “Stockfish 17.1.” Stockfish Blog, 2025\.
> >
> > \[4\] Lichess Team. “Lichess Open Database — Evaluations.” 2025\.
> >
> > \[5\] Lichess Team. “Lichess Open Database — Puzzles.” 2025\.
> >
> > \[6\] Rein et al. “GPQA: A graduate‑level, google‑proof Q\&A benchmark,” 2024\.
> >
> > \[7\] Hendrycks, Dan, et al. "Measuring mathematical problem solving with the math dataset." arXiv preprint arXiv:2103.03874 (2021).

---

> > ### Comment · Reviewer_mMdF · 2025-11-28
> > **Response to authors**
> >
> > Thank authors for the clarification.
> >
> > From the LLM perspective in 2025, the core question is no longer whether we can build a strong chess-playing agent, since AI has surpassed human in the boardgame, like AlphaGo, though it is not an LLM. The author should further show how Chess playing contributes to general intelligence and show why chess playing matters for LLM. Even though there is no new experiment, the paper should at least articulate and probe a plausible link between chess understanding and the kinds of reasoning or planning behaviors we care about outside the chess domain with existing benchmarks.
> >
> > Therefore, the authors did not solve my main concern.

---

> > > ### Author Response · Authors · 2025-12-03
> > >
> > > Thank you for your continued engagement\! We are glad to discuss the significance of ChessQA further, “from the LLM perspective in 2025”.
> > >
> > > First of all, we would like to expand on the distinction between chess **playing** and chess **understanding**. Admittedly, AI such as AlphaGo has surpassed humans in chess playing. However, building strong chess-playing agents has not yet solved the broader problems of chess understanding, teaching, and explanation. In particular, chess-playing agents only output final moves without verbalizing their *thinking* process to reach such decisions. Such clear chess understanding and explanation capabilities are the prerequisite for downstream tasks that are in high demand, such as chess tutoring, chess tournament streaming, chess puzzle recommendation, etc. LLMs hold significant potential to achieve these goals, while we haven’t fully understood how current and future LLMs perform on chess understanding, which is why ChessQA is needed in practice.
> > >
> > > Secondly, we emphasize that ChessQA focuses on both chess **knowledge** and chess **reasoning**; being chess-specific does not conflict with evaluating the long-term reasoning capabilities of LLMs. Analogously, math-specific datasets like MATH \[7\] have been widely used for evaluating LLM reasoning abilities beyond math knowledge. Chess has several desirable properties as a domain for assessing LLM capabilities. It is a neatly circumscribed, perfect-information, and fully objective domain with clear states, actions, goals, and rewards. Despite this, it is incredibly deep and admits a huge variety of skill levels. People are also still very active in playing chess, and a near-infinite supply of human chess data is available online. If we consider the Position Judgement task, even though LLM may already have chess **knowledge**, calculating centipawn advantage requires long-term planning and implicit search, which necessitates **reasoning** beyond simple chess knowledge.
> > >
> > > Third, even in the LLM stage now, chess AI itself is still an active and highly impactful research domain. Representative directions include human behavior modeling in chess \[1\] \[2\], chess-playing without search \[3\], chess interpretability \[4\], reasoning with internal representations \[5\], and chess puzzle creativity \[6\].
> > >
> > > **References**
> > >
> > > \[1\] McIlroy‑Young, R. et al. “Aligning Superhuman AI with Human Behavior: Chess as a Model System.” *KDD* 2020\.
> > >
> > > \[2\] Tang, Z. et al. “Maia‑2: A Unified Model for Human‑AI Alignment in Chess.” *NeurIPS* 2024\.
> > >
> > > \[3\] Ruoss, Anian, et al. "Amortized planning with large-scale transformers: A case study on chess." *NeurIPS*, 2024\.
> > >
> > > \[4\] McGrath, Thomas, et al. "Acquisition of chess knowledge in AlphaZero." *PNAS 2022\.*
> > >
> > > \[5\] Schultz, J. et al. “Mastering Board Games by External and Internal Planning with Language Models.” *ICML 2025*
> > >
> > > \[6\] Feng, Xidong, et al. "Generating Creative Chess Puzzles." *NeruIPS 2025\.*
> > >
> > > \[7\] Hendrycks, Dan, et al. "Measuring Mathematical Problem Solving With the MATH Dataset." *NeurIPS 2021\.*

---

### Official Review · Reviewer_b4uy · 2025-11-01

**Soundness:** 3
**Presentation:** 3
**Contribution:** 2
**Rating:** 6
**Confidence:** 3

**Summary:**

This paper introduces ChessQA, a testing and benchmarking suite for evaluating the chess understanding of LLMs across five skill categories: Structural, Motifs, Short Tactics, Position Judgment, and Semantic. The benchmark encompasses a diverse 3,500-item, 50-task set targeting chess knowledge from rules comprehension to high-level semantic commentary. The authors systematically analyze and compare the performance of 13 recent LLMs  on this benchmark, presenting results by task and model family, as well as error analyses and insights into reasoning and token efficiency.

**Strengths:**

1. The construction of ChessQA is described clearly, with detailed specifications, prompt formatting, canonicalization rules, and task generation algorithms. The authors also provide reproducibility details and plan for public release.
2. The authors conducted extensive and comprehensive evaluations and experiments.
3. The authors provided detailed error analysis, which brings more insights beyond just accuracy.

**Weaknesses:**

1. As there are not enough examples provided in the paper this raises further concerns regarding the format and quality of the data.
2.  The literature review misses several recent relevant papers that are directly pertinent to the goals and context of ChessQA, especially those benchmarking LLMs in chess and grid-based games, evaluating state tracking [1,2]



[1] Kuo, Mu-Tien et al. “Large Language Models on the Chessboard: A Study on ChatGPT's Formal Language Comprehension and Complex Reasoning Skills.” ArXiv abs/2308.15118 (2023)

[2] Topsakal, Oguzhan et al. “Evaluating Large Language Models with Grid-Based Game Competitions: An Extensible LLM Benchmark and Leaderboard.” ArXiv abs/2407.07796 (2024)

**Questions:**

1. Please check the Weaknesses.

---

> ### Author Response · Authors · 2025-11-24
> **Response to Weaknesses**
>
> Thank you for the constructive suggestions and positive feedback. We address your two concerns below and indicate exactly where we revised the paper.
>
> **W1:** We have added a section in the appendix titled “Representative Items” (Appendix D), which contains a fully formatted example for each of the categories, including the exact prompt, context, chessboard figure, and ground truth. We now explicitly reference this appendix from Section 3 immediately after the introduction to each category, so readers can quickly jump from the conceptual description to end‑to‑end examples that make the format, difficulty, and uniqueness of the data transparent. Construction details, generators, and canonicalization rules remain in Appendix C, which already specifies input/output types and exact‑match scoring for all tasks. Together, these changes keep the main text focused while directly addressing your concern about format clarity and data quality.
>
> **W2:** We have added the mentioned related works into the revised paper. In the revised Section 2 (“Chess‑specific LLM evaluations”) and Table 1, we additionally include \[1\], highlighting their focus on formal encodings and localized board‑state reasoning, and we explain how ChessQA complements this line by unifying rules/state tracking, motif recognition, tactics, position evaluation, and grounded commentary within a single benchmark. We also add \[2\] into the related work section, emphasizing that their competition‑style, agentic setting across multiple grid games provides a valuable gameplay perspective, whereas ChessQA deliberately isolates text‑to‑text chess competence using verifiable outputs (UCI/FEN, bucketed centipawn labels, and comment selection).
>
> **References**
>
> \[1\] Kuo, M.-T., Hsueh, C.-C., & Tsai, R. T.-H. (2023). *Large Language Models on the Chessboard: A Study on ChatGPT’s Formal Language Comprehension and Complex Reasoning Skills*. arXiv:2308.15118.
>
> \[2\] Topsakal, O., Edell, C. J., & Harper, J. B. (2024). *Evaluating Large Language Models with Grid-Based Game Competitions: An Extensible LLM Benchmark and Leaderboard*. arXiv:2407.07796.

---

### Official Review · Reviewer_qH3B · 2025-11-01

**Soundness:** 3
**Presentation:** 2
**Contribution:** 2
**Rating:** 4
**Confidence:** 2

**Summary:**

The paper introduces ChessQA, a comprehensive QA-style benchmark to evaluate LLMs’ chess understanding across five progressively abstract categories: Structural (rules/state), Motifs (pattern recognition), Short Tactics (single-move puzzle solving), Position Judgment (centipawn evaluation classification), and Semantic (selecting the best human commentary for a given position+move among various distractors). The benchmark comprises 50 tasks with 3,500 items. Several contemporary LLMs are evaluated finding that: explicit reasoning markedly improves accuracy (~+14.7 points) but consumes many more tokens; models excel at Structural tasks (best ~97%) yet struggle on Short Tactics (mean ~17%) and Position Judgment (mostly near chance). Frequent failure modes are reported as well which are as follows: board-state hallucination, legality errors, sound analysis but wrong final action, and false “no answer”, and propose ChessQA as a dynamic benchmark because of task difficulty calibration with a public leaderboard and periodic refreshes.

**Strengths:**

Strengths:

1. Clear task categorization mirroring human learning trajectories: Standardized notations (FEN/PGN/UCI) along with the five-category curriculum mirrors typical human learning trajectories-from rules to explanation, yielding a richer, multi-dimensional assessment than single “best-move” tasks helping assess where modern llms lack in terms of capabilities
2. Deterministic scoring: The benchmark leans on exact-match outputs, canonicalization (e.g., sorted sets for legal moves), and engine-backed centipawn labels; motif detectors use ray scans/legality simulation; Semantic tasks enforce single correct comment with structured distractors. All these enforce determinism in scoring mechanism eliminating any randomness that could be induced because of the scoring mechanisms
3. Dynamic, parameterized construction: Ensuring difficulty is calibrated via data selection (puzzle ratings, cp buckets) and option/distractor design (tight cp grids; keyword/piece-stage/embedding retrieval) makes the dataset to evolve ensuring it won't become obsolete
4. Actionable error analysis: The four failure modes are concrete helping in understanding where the current models lack and focus on those tasks

**Weaknesses:**

Weaknesses:

1. No proper backing for fixed five-option mapping: Mapping engine cp to the fixed five-option grid {−400, −200, 0, 200, 400} is coarse and unclear why these boundaries serve well compared against any other setting. For example, a seven-option grid or a three-option grid or different values in the five-option grid itself
2. No Depth-Based Tactical Reasoning Analysis: The benchmark does not assess whether models can think/think how many moves ahead and make tactical sacrifices. These deeper calculations and thinking several moves ahead are essential for real-world chess strength, where short-term material loss can lead to long-term positional or strategic gains and models thinking several moves ahead get the advantage. Without this dimension, the evaluation overlooks a critical component of advanced chess reasoning
3. Accuracy vs token analysis is not very comprehensive: The thinking and non-thinking modes reveal stark differences but not clear of how much amount of thinking (low v medium v high) and controlled temperature affects accuracy
4. Potential training-data leakage: LLM pretraining data might almost certainly includes FEN/PGN/UCI and popular puzzle/game corpora. The paper does not quantify against potential data leakage which could be the reason why it scored high on some tasks

**Questions:**

Check the weakness section

---

> ### Author Response · Authors · 2025-11-24
> **Response to Weaknesses**
>
> We thank the reviewer for the thoughtful review and for highlighting the strengths of ChessQA’s structure, deterministic scoring, and error analysis. Below we respond to each of the weaknesses and clarify how the revised manuscript addresses them. All section and appendix references are to the updated version of the paper.
>
> **W1:** Any multiple‑choice formulation necessarily imposes discrete buckets on a continuous scale, so some choice of thresholds is unavoidable. We adopt the five‑option grid {−400, −200, 0, 200, 400} as a reasonable instantiation out of many. In particular, these values are aligned with common practice in human chess annotation: “clearly worse”(-400), “slightly worse”(-200), “equal”(0), “slightly better”(200), and “clearly better”(400) correspond to standard Numeric Annotation Glyph categories (e.g., “±”, “+/−”) in PGN \[1\], one of the most commonly used chess notation. Modern engines expose centipawn scores that chess practitioners routinely map to these discrete bands when annotating games. For example, Stockfish’s NNUE evaluation \[2\] and the Stockfish 17.1 release notes \[3\] are representative examples of such usage, as is the Lichess Evaluations database \[4,5\], which aggregates engine scores for millions of positions that players mentally convert to corresponding qualitative labels in practice. In Section 3.1, we clarify that our pipeline already supports alternative grids; for example, tightening the buckets to {−200, −100, 0, 100, 200} is described as a built‑in option for raising task difficulty. In future leaderboard releases, we plan to expose additional 3 and 7‑option variants so that the community can empirically probe sensitivity to this design choice. We have added the abovementioned support for option construction in Section 3.3.
>
> **W2:** First, we explicitly enforce multi-move reasoning in state‑tracking tasks under the Structural category. Specifically, an initial FEN and a sequence of UCI moves are provided for LLMs to infer the exact terminal state after applying these moves, which directly tests long‑horizon legality reasoning. Please refer to Table 2 and Section C.1 in the Appendix for more details. Second, many sub-tasks in our benchmark implicitly enforce long-term look-ahead and deeper calculations. While we do not ask models to emit full principal variations, long-term calculations appear to be prerequisite to solving these tasks. Examples include Short Tactics and Position Judgement tasks, where solving the puzzle or evaluating the position almost always requires anticipating likely continuations and, in many cases, tactical sacrifices rather than spotting a one‑move motif. In particular, as described in Section 3.3, the Position Judgement category is specifically designed for long-term calculation and evaluation.
>
> **W3:** We have run more experiments beyond the base‑vs‑thinking comparisons: We use different settings of thinking budgets on a representative thinking‑enabled model, i.e., Claude Haiku‑4.5, and a temperature sensitivity test for its non‑thinking variant. Accuracy rises monotonically as we raise the reasoning effort from minimal/low to medium and high (28.9%, 49.4%, and 54.0%, respectively); item‑paired t‑tests over 3 independent runs per setting indicate the low-to-medium jump is highly significant and the medium-to-high improvement is smaller but still significant. On the other hand, lowering the temperature from 1.0 to 0.7 yields no statistically significant difference overall. We summarize these findings in Section 4.2 and provide full per‑category results, significance tests, and figures in Appendix B, complementing the current Tables 5, 6, and 7\. Due to a limited budget, we are unable to run the above experiments for every model, which may necessitate community efforts.

---

> > ### Author Response · Authors · 2025-11-24
> > **Supplement to 'Response to Weaknesses'**
> >
> > **W4:** We agree that pre‑exposure to chess data is an important consideration, and our benchmark prevents data contamination from three aspects. First, we use proprietary source data: part of our source data is from ChessBase 17, which is a licensed and paid database that is unlikely to appear in open web‑scale training corpora \[6\]. Second, we use refreshing source data: part of our source data is from Lichess puzzles and evaluations databases, which are continually updated monthly. This allows us to use “fresh” slices of data that post‑date LLMs’ training cutoff \[4,5\]. Specifically, we use the latest possible positions in Lichess database at the time of constructing ChessQA. Third, we transform raw chess games into QA formats with customized functions, ensuring our tested patterns do not exist in the open online sources repeatedly. Even if a model had seen a particular PGN or FEN during pre‑training, it would still have to understand the position and carry out reasoning to answer the exact question. We have updated Section 3.2 to reflect the above discussion.
> >
> > **References**
> >
> > \[1\] Edwards, S. J. (1994). *Portable Game Notation (PGN) Specification and Implementation Guide.*
> >
> > \[2\] Stockfish Developers. “Introducing NNUE Evaluation.” *Stockfish Blog*, 2020\.
> >
> > \[3\] Stockfish Developers. “Stockfish 17.1.” *Stockfish Blog*, 2025\.
> >
> > \[4\] Lichess Team. “Lichess Open Database — Evaluations.” 2025\.
> >
> > \[5\] Lichess Team. “Lichess Open Database — Puzzles.” 2025\.
> >
> > \[6\] ChessBase GmbH. *ChessBase 17: Commenting Functions (Text & Variants).* 2023\.

---

### Author Response · Authors · 2025-12-03

#### We sincerely thank all reviewers for their constructive feedback and engaging discussions throughout the review period. Below we summarize the main concerns raised, how we addressed them, and the improvements made in the revision.

#### Reviewers recognized several key strengths of ChessQA: a rigorous and useful complement to existing chess benchmarks, with the ability to understand model failure modes being a convincing benefit even as model performance increases (Reviewer 6T1y); clear task categorization mirroring human learning trajectories, with standardized notations (FEN/PGN/UCI) and a five-category curriculum yielding a richer, multi-dimensional assessment than single "best-move" tasks (Reviewer qH3B); deterministic scoring through exact-match outputs, canonicalization, and engine-backed labels that eliminate randomness (Reviewer qH3B); dynamic, parameterized construction allowing the dataset to evolve (Reviewer qH3B); actionable error analysis with concrete failure modes that help identify where current models lack capabilities (Reviewer qH3B); and broad coverage spanning 3,500 examples across 50 task types with evaluation of a wide range of models (Reviewer mMdF).

#### **Motivation and Comparison with Game-Arena Benchmarks (Reviewer 6T1y)**

#### Reviewer 6T1y raised an important question about the necessity of ChessQA given existing game-arena evaluations. Through our discussion, we clarified that ChessQA complements rather than replaces gameplay benchmarks. While game-arena Elo provides an all-in-one metric, it collapses many distinct failure modes into a single coarse outcome. ChessQA decomposes chess ability into five interpretable sub-skills that map closely to skills humans learn as they improve. This decomposition is especially valuable for chess understanding and education applications: a system that reliably explains motifs and evaluations may be more pedagogically valuable than one whose raw Elo is slightly higher but whose reasoning is opaque. ChessQA directly evaluates the understanding and explanation capabilities that are prerequisites for downstream tasks in high demand, such as chess tutoring, chess tournament streaming, and chess puzzle recommendation. Following the reviewer's suggestion, we added a direct comparison between GameArena and ChessQA rankings for overlapping models in the revision, showing a clear correlation while also revealing that the two benchmarks capture different aspects of chess understanding.

#### **Data Contamination Concerns (Reviewers qH3B, 6T1y)**

#### Multiple reviewers asked about potential training-data leakage. We addressed this from two aspects: (1) refreshing source data from Lichess databases that post-date LLMs' training cutoffs, and (2) transformation of raw chess games into QA formats that do not exist in open online sources. Following Reviewer 6T1y's suggestion for empirical validation, we conducted a statistical analysis comparing model performance on puzzles dated before and after the training cutoff. Using Claude Haiku 4.5 on Short Tactics, we found no statistically significant difference between pre-cutoff (117/900) and post-cutoff (113/900) puzzles (z-test p=0.78, chi-squared p=0.83, Fisher's exact p=0.83), providing empirical evidence that models did not benefit from potential exposure to older puzzles during pre-training, and therefore that data contamination did not substantively affect our results.

#### **Position Judgment Options (Reviewers qH3B, mMdF)**

#### Two reviewers questioned the justification for the five-option centipawn grid {−400, −200, 0, 200, 400}. First, we point out that the choice of centipawn values to use is necessarily arbitrary. Given this, we clarified that our choice of values is principled as they align with common practice in human chess annotation, corresponding to standard Numeric Annotation Glyph categories in PGN (e.g. “White is winning”, “White is better”, “Equal”, etc.). We also noted that our pipeline already supports alternative grids (e.g., {−200, −100, 0, 100, 200} for higher difficulty), and we plan to expose 3 and 7-option variants in future leaderboard releases. Altering the centipawn grid provides us with a clean way to calibrate the difficulty of the questions.

---

> ### Author Response · Authors · 2025-12-03
>
> #### **Chess Knowledge and Long-term Reasoning (Reviewers qH3B, mMdF)**
>
> #### Reviewer qH3B noted the absence of explicit depth-based tactical analysis. We clarified that multi-move reasoning is already enforced in our state-tracking tasks under the Structural category, where models must infer terminal states after applying move sequences. Additionally, Short Tactics and Position Judgment implicitly require long-term look-ahead, as solving puzzles or evaluating positions almost always requires anticipating continuations and variation rather than spotting one-move motifs.
>
> #### Moreover, we emphasize that ChessQA focuses on both chess **knowledge** and chess **reasoning**; being chess-specific does not conflict with evaluating the long-term reasoning capabilities of LLMs. Analogously, math-specific benchmarks have been widely used for evaluating LLM reasoning abilities beyond math knowledge. Chess has several desirable properties as a domain for assessing LLM capabilities. It is a neatly circumscribed, perfect-information, and fully objective domain with clear states, actions, goals, and rewards. Despite this, it is incredibly deep and admits a huge variety of skill levels. People are also still very active in playing chess, and a near-infinite supply of human chess data is available online. If we consider the Position Judgement task, even though LLM may already have chess knowledge, calculating centipawn advantage requires long-term planning and implicit search, which necessitates reasoning beyond simple chess knowledge.
>
> #### **Reasoning Budget Analysis (Reviewer qH3B)**
>
> #### We conducted additional experiments with different thinking budgets on Claude Haiku 4.5. Accuracy rises monotonically from minimal/low (28.9%) to medium (49.4%) to high (54.0%), with item-paired t-tests over 3 independent runs confirming statistical significance. Temperature sensitivity tests showed no significant difference when lowering temperature from 1.0 to 0.7. This is a nice bonus result that we have included and strengthens the paper, and we thank the reviewer for this suggestion.
>
> #### **Chess Understanding and Chess Playing (Reviewer mMdF)**
>
> #### Reviewer mMdF inquired about the connection between chess understanding and general intelligence. Through our discussion, we emphasized the distinction between chess playing and chess understanding. While AI has surpassed humans in gameplay, building strong chess-playing agents has not solved the broader problems of chess understanding. Traditional agents output final moves without verbalizing their reasoning, limiting their utility for chess education, commentary, and human behavior modeling. LLMs hold significant potential to achieve these downstream tasks, motivating the need for ChessQA to evaluate how current and future LLMs perform on chess understanding. Importantly, chess as a model system has proven valuable for AI research beyond gameplay itself. Recent work from DeepMind and other leading research groups has leveraged chess to study fundamental questions, including how AI systems acquire knowledge and concepts \[4\], whether transformers can perform planning without explicit search \[3\], and how to align superhuman AI with human behavior \[1,2\]. Chess provides a neatly circumscribed, perfect-information, and fully objective domain with clear states, actions, goals, and rewards, making it an ideal testbed for probing these capabilities. ChessQA contributes to this broader research agenda by providing a systematic evaluation framework for chess understanding in LLMs.
>
> #### **Benchmark Saturation Concerns (Reviewer 6T1y)**
>
> #### To address whether ChessQA will soon be saturated, we constructed an additional test set of 100 hard Lichess puzzles with ELO ratings over 3000\. GPT-5 with high reasoning effort solved only 26/100 puzzles (averaging \~24k tokens per puzzle), confirming that expert-level tactics remain challenging even for the best model we tested. Moreover, ChessQA has two key properties that ensure long-term relevance: first, our source data from Lichess databases is continually updated monthly, allowing us to use fresh slices of data that post-date LLMs' training cutoffs; second, chess as a domain admits a huge variety of skill levels, and our pipeline supports flexible difficulty calibration via data selection (e.g., puzzle ratings, centipawn buckets) and option/distractor design. These properties enable ChessQA to evolve alongside frontier models and remain informative as they undergo further improvements.

---

> > ### Author Response · Authors · 2025-12-03
> >
> > **Summary of Revisions**
> >
> > We have made substantial updates to the manuscript in response to reviewer feedback:
> >
> > 1. Added discussion at the end of Section 2 clarifying how ChessQA complements game-arena evaluations.
> > 2. Updated Section 3.2 to describe our contamination prevention strategies, and conducted new empirical analysis comparing model performance on pre-cutoff vs. post-cutoff puzzles with statistical tests (z-test, chi-squared, Fisher's exact).
> > 3. Expanded Section 3.3 to justify the five-option centipawn grid with references to standard PGN annotation practices, and clarified that our pipeline supports alternative grids for future leaderboard variants.
> > 4. Added new experiments in Appendix B.2 reporting accuracy across minimal/low, medium, and high thinking budgets on Claude Haiku 4.5, with statistical significance tests over 3 independent runs. Temperature sensitivity results added in Appendix B.3.
> > 5. Added results on a new hard puzzle test set (100 Lichess puzzles with ELO \> 3000), demonstrating that GPT-5 with high reasoning effort solved only 26/100.
> > 6. Added representative examples for each category in Appendix D, with explicit references from Section 3\.
> > 7. Incorporated additional related works in Section 2 and Table 1\.
> > 8. Included binomial standard errors in Figure 3\.
> > 9. Added a Limitations section (Section 6).
> > 10. Fixed typos and clarified terminology (e.g., GPT-5 naming conventions in Section 4.1).
> >
> > We believe these revisions substantially strengthen the paper and address the reviewers' main concerns. We thank the reviewers again for their time and valuable feedback.

---

### Meta-Review · Area_Chair_TQgp · 2026-01-03

**Summary:**

This paper introduces **ChessQA**, a QA-style benchmark for evaluating chess understanding in large language models across five categories spanning rules, motifs, tactics, position evaluation, and semantic commentary. The benchmark is carefully constructed and evaluated on a wide range of contemporary LLMs.

A central concern is **the significance and positioning of the contribution**. While ChessQA is broader than many prior chess benchmarks, reviewers consistently view it as incremental relative to existing chess and grid-based evaluations, particularly gameplay-based benchmarks that already assess end-to-end chess competence. Several reviewers questioned whether a QA-based formulation provides fundamentally new insights beyond what can be obtained from interactive gameplay evaluations, and whether the added granularity justifies a new benchmark at a major venue.

Closely related is the issue of **motivation and scope**. Multiple reviewers found the original framing—suggesting chess as a general testbed for LLM reasoning—overstated. Although the authors clarified in the rebuttal that ChessQA is intended as a chess-specific diagnostic rather than a universal reasoning benchmark, this reframing highlights a remaining concern: **the paper does not convincingly demonstrate why chess understanding, as measured by ChessQA, is especially informative for broader LLM reasoning or planning capabilities**, nor does it empirically establish strong links to non-chess reasoning benchmarks. As a result, the broader impact of the benchmark remains unclear.

Reviewers also raised **methodological and design concerns**, several of which persist despite reasonable rebuttals. These include the coarse and somewhat arbitrary discretization used in the Position Judgment tasks, the lack of a human baseline to contextualize difficulty, and the relatively small number of examples per subtask. While the authors provided justifications and additional analysis (e.g., alternative bucket options and binomial error estimates), these choices still limit the interpretability and perceived rigor of certain benchmark components.


**Recommendation:** Reject.
While ChessQA is a solid and careful benchmark contribution, its incremental nature, limited demonstrated impact beyond the chess domain, and unresolved questions about necessity relative to existing evaluations place it below the acceptance threshold for ICLR.

**Reviewer Concerns:**

As above

**Reviewer Scores:**

I do not think the reviewers would change the scores.

---

### Decision · Program_Chairs · 2026-01-26

Reject